# Analysis of cellular behavior and cytoskeletal dynamics reveal a constriction mechanism driving optic cup morphogenesis

María Nicolás-Pérez[1], Franz Kuchling[1,2], Joaquín Letelier[1], Rocío Polvillo[1], Jochen Wittbrodt[2], Juan R Martínez-Morales[1]*

[1]Centro Andaluz de Biología del Desarrollo, Seville, Spain; [2]Centre for Organismal Studies, COS, University of Heidelberg, Heidelberg, Germany

**Abstract** Contractile actomyosin networks have been shown to power tissue morphogenesis. Although the basic cellular machinery generating mechanical tension appears largely conserved, tensions propagate in unique ways within each tissue. Here we use the vertebrate eye as a paradigm to investigate how tensions are generated and transmitted during the folding of a neuroepithelial layer. We record membrane pulsatile behavior and actomyosin dynamics during zebrafish optic cup morphogenesis by live imaging. We show that retinal neuroblasts undergo fast oscillations and that myosin condensation correlates with episodic contractions that progressively reduce basal feet area. Interference with *lamc1* function impairs basal contractility and optic cup folding. Mapping of tensile forces by laser cutting uncover a developmental window in which local ablations trigger the displacement of the entire tissue. Our work shows that optic cup morphogenesis is driven by a constriction mechanism and indicates that supra-cellular transmission of mechanical tension depends on ECM attachment.

*For correspondence: jrmarmor@upo.es

Competing interests: The authors declare that no competing interests exist.

## Introduction

The shape of animal organs evolved by natural selection under constrains imposed both by organ physiology in the adult and tissue mechanics during embryogenesis. Throughout development, genetic programs coordinate the behavior of single cells allowing the self-assembly of coherent tissues and tridimensional organs. Regardless of the nature of the process (i.e. either cell migration, epithelial bending or cell intercalation), mechanical tensions need to be transmitted at a supra-cellular scale for organ morphogenesis to occur. Mechanical forces, however, are generated by the contractile cytoskeleton of the constituent cells of a tissue (*Mammoto et al., 2013*; *Heisenberg and Bellaiche, 2013*). The main force generator during morphogenesis results from the molecular interaction between myosin II motors and the actin filaments at the cellular cortex (*Salbreux et al., 2012*). This actomyosin contractile apparatus sustains cortical tension, pulling cells into shape during development and tissue homeostasis. Contractile forces are then transmitted to neighboring cells and to the extracellular matrix (ECM) through cadherin and integrin receptors, allowing individual cell contributions to be integrated into tensions at the tissue/organ level (*Papusheva and Heisenberg, 2010*; *Lecuit et al., 2011*). Regardless the morphogenetic context, actomyosin contractile forces are resisted both by cellular adhesions and by the compression of the internal cytoskeleton itself. This results in a balance of forces that stabilizes transiently cell and tissue shapes for each stage of the developmental program that builds up a given organ.

**eLife digest** Tissues and organs form into their final shapes because the cells in a developing embryo generate forces that alter their shape and position. Networks of fibres made from actin and myosin proteins generate these forces, and because the fibres can assemble in many different ways inside cells, they allow the cells to move and change shape in many different ways.

Forces in some tissues can cause flat sheets of cells to bend. These sheets of cells are attached on one side (their "basal" surface) to a collection of membranes and molecules that are known as the extracellular matrix. When the cells in the sheet progressively shrink at their basal surface, causing the sheet to bend towards the extracellular matrix, this is known as basal constriction.

Nicolás-Pérez et al. have used high-resolution imaging to record how basal constriction helps the optic cup – the main chamber of the eye – to form in zebrafish embryos. This imaging confirmed that a sheet of precursor cells progressively bends towards its basal surface to form the curved shape of the eyeball. Further analysis revealed that this basal constriction happens when myosin fibres accumulate in clusters along the basal surface of some of the precursor cells. The resulting contraction of the basal surface of the cells relies both on the tension generated by myosin inside the cell and on the cells being attached properly to the extracellular matrix.

Using a laser beam, Nicolás-Pérez et al. also destroyed small parts of the basal surface of the retina. This procedure allows the mechanical tension distribution throughout the developing eye to be mapped. Laser ablations revealed a narrow time window during development when destroying small parts of the basal surface can cause the entire sheet of cells to relax, preventing it from curving to form the shape of the eye.

Sheets of precursor cells are important building blocks of the nervous system, yet researchers only have limited knowledge of the processes that enable them to fold or bend into a final shape. As such, the findings of Nicolás-Pérez et al. will contribute to a wider understanding of how cells and tissues behave while the brain is forming.

Live-imaging studies have examined actomyosin architecture and dynamics in different morphogenetic models. The emerging picture reveals a wide variety of cortical actomyosin behaviors and localizations depending on the tissue context. Initial reports, focused in epithelial constriction processes, revealed pulsatile myosin flows preceding the periodic contraction of the cellular cortex. This has been reported in *Drosophila* epithelia either at the apical cortex, during mesoderm invagination or germ-band extension (*Martin et al., 2009*; *Gorfinkiel and Blanchard, 2011*; *Roh-johnson et al., 2012*; *Rauzi et al., 2010*), or at the basal surface during egg chamber elongation (*He et al., 2010*). Oscillatory actomyosin flows can be coupled to the stabilization of the cells in a 'constricted' state after each pulse, thus resulting in a progressive (i.e. ratcheted) reduction of the cellular apex (*Martin et al., 2009*; *Rauzi et al., 2010*). Alternatively, the cell cortex may oscillate, contracting and relaxing, without a net reduction of the area over time (*He et al., 2010*; *Solon et al., 2009*). Furthermore, actomyosin flows may direct epithelial morphogenesis operating in a continuous non-pulsatile manner, as described during zebrafish epiboly (*Behrndt et al., 2012*). Notably, the actomyosin network localizes in circumferential (i.e. junctional) belts in the vertebrate neural tube (*Nishimura et al., 2012*), instead of medio-apically as observed in several *Drosophila* epithelia (*Gorfinkiel and Blanchard, 2011*; *Martin et al., 2009*) and in gastrulating cells in *Xenopus* (*Kim and Davidson, 2011*). In the context of the current study, although actomyosin distribution has been analyzed during optic cup morphogenesis in vertebrates (*Chauhan et al., 2009*; *Martinez-morales et al., 2009*), its dynamics has not been examined in vivo.

Vertebrate eye development has been a common subject of interest for classical embryologists as well as modern developmental geneticists (*Spemann, 1901*; *Fuhrmann, 2010*; *Sinn and Wittbrodt, 2013*). The process entails first the protrusion of the eye progenitors to form the lateral optic vesicles, and subsequently the infolding of this tissue into bi-layered optic cups (*Li et al., 2000*; *Schmitt and Dowling, 1994*; *Hilfer, 1983*; *Schook, 1980*). Live imaging followed by cell tracking of retinal progenitors in zebrafish revealed that optic vesicle bulging is driven by the rearrangement and epithelialization of individual cells (*Brown et al., 2010*; *Rembold et al.,*

*2006*; *England et al., 2006*; *Ivanovitch et al., 2013*). In contrast to teleosts, in amniotes and cartilaginous fishes optic vesicles develop by epithelial folding from an already hollow neural tube (*Lowery and Sive, 2004*). The morphogenesis of the vertebrate optic cup has also been examined in live imaging studies, both in teleost models (*Kwan et al., 2012*; *Martinez-morales et al., 2009*; *Picker et al., 2009*; *Heermann et al., 2015*), as well as in self-organized organs from ES-cultured cells in mammals (*Nakano et al., 2012*; *Eiraku et al., 2011*). Although optic cup formation seems less divergent among vertebrates than vesicles' evagination, some particularities in cell behavior have been observed and different mechanisms proposed. In mouse embryos, contractile filopodia connecting neural retina and lens epithelia have been shown to adjust the final curvature of both epithelia (*Chauhan et al., 2009*). However, optic cup development can be recapitulated in vitro in ES cells aggregates suggesting that the morphogenetic program is to a large extent intrinsic. Using this in vitro model, it has been hypothesized that optic cup invagination is driven by the apical constriction of the neuroepithelial cells located at the rim between the presumptive retina and RPE domains (*Eiraku et al., 2011*, *2012*). Tracking of individual cells in zebrafish has shown that epithelial flow through this rim contributes to neural retina expansion (i.e. at the expenses of the RPE) and optic cup folding (*Heermann et al., 2015*; *Kwan et al., 2012*; *Picker et al., 2009*). Whether cell involution and apical constriction at the rim are species-specific mechanisms or operate coordinately in the same organism is still an open question. Finally, we previously postulated the basal constriction of the neuroblasts as an active mechanism contributing to optic cup morphogenesis (*Martinez-Morales et al., 2009*; *Martinez-Morales and Wittbrodt, 2009*). The polarized trafficking of integrin receptors toward the basal surface of the epithelial cells plays an essential role during retinal morphogenesis in teleosts. We showed that this process is controlled by the molecular antagonism between the trans-membrane protein opo and the clathrin adaptors numb and numb-like (*Bogdanovic et al., 2012*). In *opo* medaka mutants, basal feet appear wider and disorganized in the retina (*Martinez-morales et al., 2009*). Although this observation suggests a progressive reduction of the neuroblasts feet, the constriction process has not been formally examined in vivo.

Through quantitative imaging, here we characterize the pulsed contractile behavior of the retinal neuroblasts during optic cup folding in zebrafish. We explore actomyosin dynamics and show that accumulation of myosin foci in scattered cells is associated with contraction of the cellular feet. We show that interference with myosin II function or laminin-mediated basal attachment impairs cell contractility and affect retina folding. To further characterize this morphogenetic process at tissue level, we locally ablate the neuroepithelium to map mechanical tensions through development. This approach identified a narrow developmental window in which local ablation of the retina at its basal surface triggers the global displacement of the retinal epithelium. Our work shows that the myosin-dependent generation of constrictions forces in individual neuroblast and their transmission at a supra-cellular scale play an essential role during optic cup folding in zebrafish.

## Results

### Retinal precursors undergo basal constriction and display oscillatory contractions during optic cup folding

To formally show that basal constriction is taking place as the optic cup forms, we investigated the behavior of retinal precursors by live-imaging analysis. Retinae from the zebrafish line *tg(vsx2.2:GFP-caax)*, in which precursors' plasma membrane is uniformly labeled, were imaged through morphogenesis starting at 17 hpf (*Figure 1A–H*; *Video 1*). Tissue recordings evidenced a complete epithelial organization shortly after 17 hpf, with mitotic rounding happening apically throughout the entire folding process. In agreement with previous reports, cell involution was also observed at the rim between the RPE and neural retina, particularly from 20 hpf on and at the posterior (i.e. temporal) border of the cup (*Heermann et al., 2015*; *Kwan et al., 2012*; *Picker et al., 2009*). As morphogenesis proceeds, GFP-caax signal become brighter at the basal side in the central retina, suggesting an increased membrane density in this region. Moreover, whereas the length of the apical edge of the retina increased significantly, the basal length remained invariant (*Figure 1I*). This observation, in conjunction with the previously reported increase (1.5x) in retinal cells number within this developmental window (*Kwan et al., 2012*), suggests a progressive narrowing of the basal feet between 17 and 24 hpf. Cell elongation, a common phenomenon in many constricting epithelia (*Sawyer et al.,*

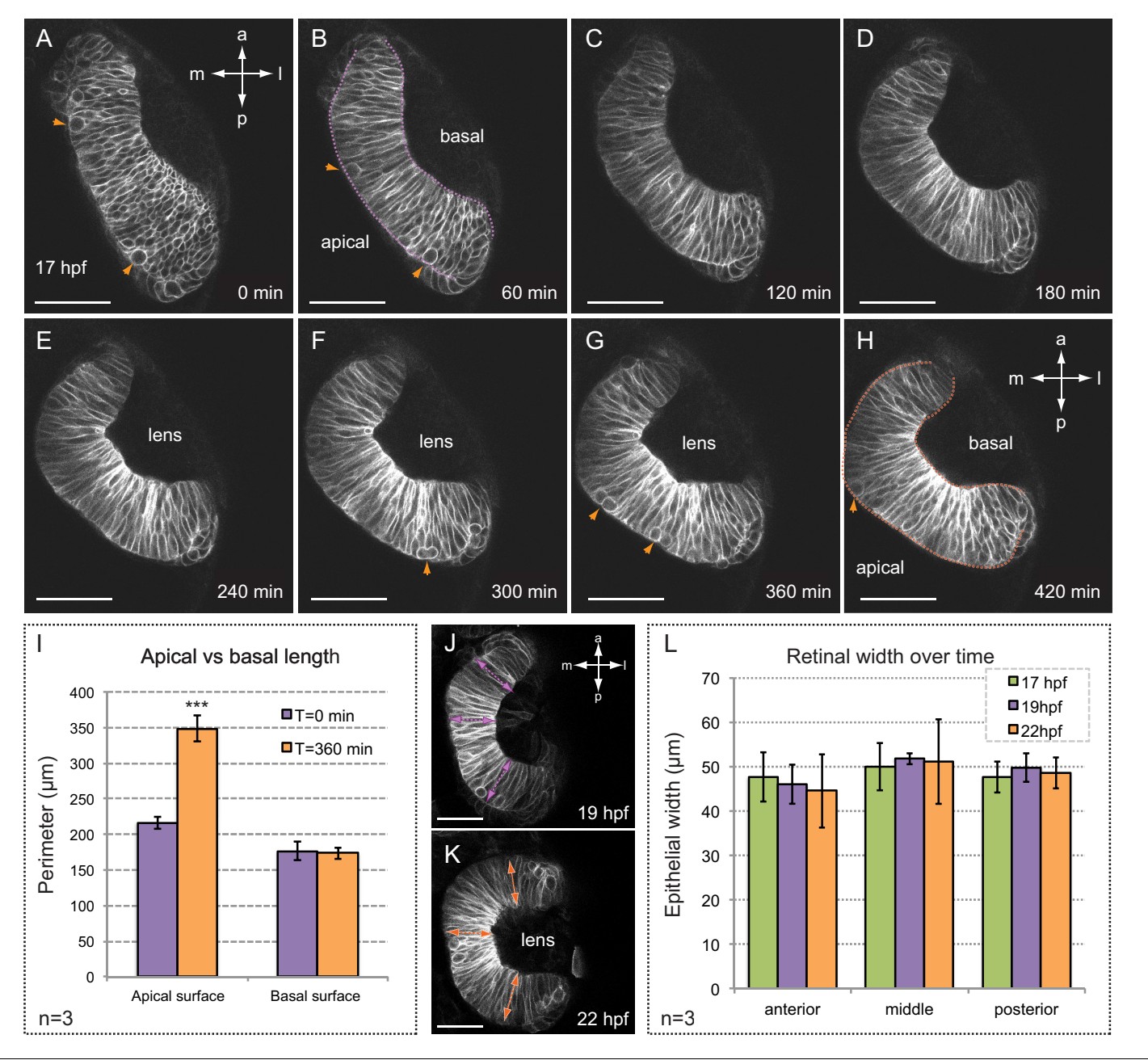

**Figure 1.** Folding of the retinal epithelium in zebrafish. (A–H) Time series of optical sections show the progression of retinal morphogenesis starting at 17 hpf (dorsal view) in a tg(vsx2.2:GFP-caax) embryo. Arrowheads point to mitotic divisions at the apical surface. Apical and basal edges are indicated at 60 (purple) and 420 (orange) min. See also *Video 1*. (I) Quantification of the perimeter of the apical and basal edges between 18 and 24 hpf. (J–L) Retinal width remains constant throughout retinal folding as revealed in tg(vsx2.2:GFP-caax) embryos. Error bars indicate s.d. of the mean. (n = 3; T-test). Antero-posterior and medio-lateral axes are indicated. Scale bars = 50 μm.

The following figure supplements are available for figure 1:

**Figure supplement 1.** Imaging setup and segmentation.

**Figure supplement 2.** Neuroblasts' area quantification during eye morphogenesis.

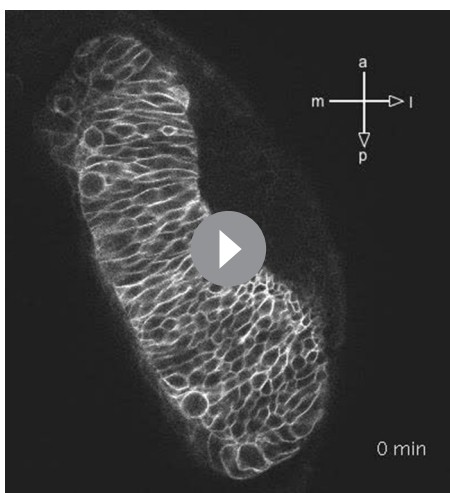

**Video 1.** Time lapse of zebrafish optic cup folding. Optical section from a *tg(vsx2.2:GFP-caax)* embryo showing the folding of the retinal tissue. Imaging starts at 17 hpf. Antero-posterior and medio-lateral axes are indicated. See also *Figure 1*.

*2010*), does not occur during retinal folding, as the width of the tissue remained constant ($\approx$ 50 µm) through the process (*Figure 1J–L*).

To investigate directly the constriction process, we examined the dynamics of both apical and basal neuroblasts' surfaces within the most critical morphogenetic window, between 19 and 21 hpf, using again the *tg(vsx2.2:GFP-caax)* line. Processed images were segmented and individual cell areas tracked through time (*Figure 1—figure supplement 1*). During this developmental window, basal areas shrank significantly (40%) and progressively from 25.4 ± 1.7 to 15.3 ± 1.5 µm$^2$ (n = 24). Maximum basal constriction was observed between 19 and 20 hpf when most of the cells significantly reduced their area in a 30 min period (74.2% and 66.7% respectively; *Figure 1—figure supplement 2*). Interestingly, this developmental window coincides with the acute bending of the retinal epithelium (*Figure 1*). In contrast, apical areas remained constant between 19 and 20 hpf and even expanded (28%) at later stages, between 20 and 21 hpf (*Figure 1—figure supplement 2*).

Live imaging analyses revealed periodic contractions occurring at apical and basal cell surfaces (*Video 2*, *Figure 2*), which may resemble the pulsatile behavior observed in constricting epithelia in both vertebrate and invertebrate tissues (*Martin et al., 2009; Solon et al., 2009; Rauzi et al., 2010; He et al., 2010; Kim et al., 2011*). As previously reported for *Drosophila* epithelia (*Martin et al., 2009*), analysis of pulsed contractions in adjacent retinal cells revealed that these are mostly asynchronous (*Figure 2—figure supplement 1*). The analysis of individual cells from three independent retinas showed that 76% of the apical (n = 43) and 90% of the basal (n = 46) oscillations presented no major correlation with those of their neighbors (Pearson correlation coefficient R < | 0.5|). Comparison of the pulsatile behavior at both epithelial planes revealed significant differences. Although both surfaces oscillate with a similar frequency of 50 ± 12.5 *mHz* ($\approx$ 20 ± 5 s; n = 26 cells), the peak-to-peak amplitude is considerably larger at the basal 11.1 ± 1.3 µm$^2$/min than at the apical surface 4.1 ± 0.57 µm$^2$/min (*Figure 2—figure supplement 1*). Of note, whereas a progressive reduction of cell area was apparent at the basal side, cells did not display a net constriction at the apical side over a 25-min period (*Figure 2*). This

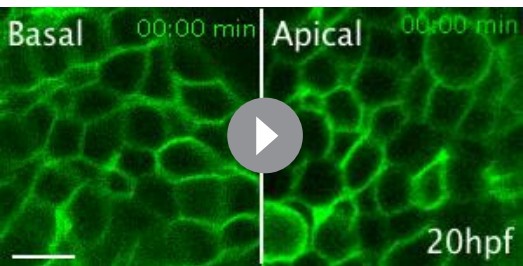

**Video 2.** Membrane oscillations at the basal and apical surfaces. Maximum projection of 3 z-stacks (over a total of 1 µm) at the basal and apical surfaces in a *tg(vsx2.2:GFP-caax)* retina show the oscillatory behavior of the cell membranes over a period of 35 min. Images were acquired every 5 s. Scale bars = 10 µm. See also *Figure 2*.

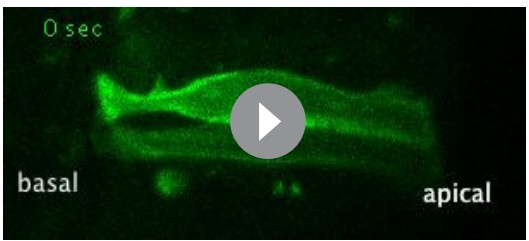

**Video 3.** Analysis of *tg(vsx2.2:GFP-caax)* clones show uncoupled oscillations at apical and basal surfaces. Maximum projection of 3 z-stacks (over a total of 1 µm) along the apico-basal axis shows the oscillatory behavior of apical and basal edges simultaneously in *tg(vsx2.2:GFP-caax)* clones. Images were acquired every 8 s. See also *Figure 3*.

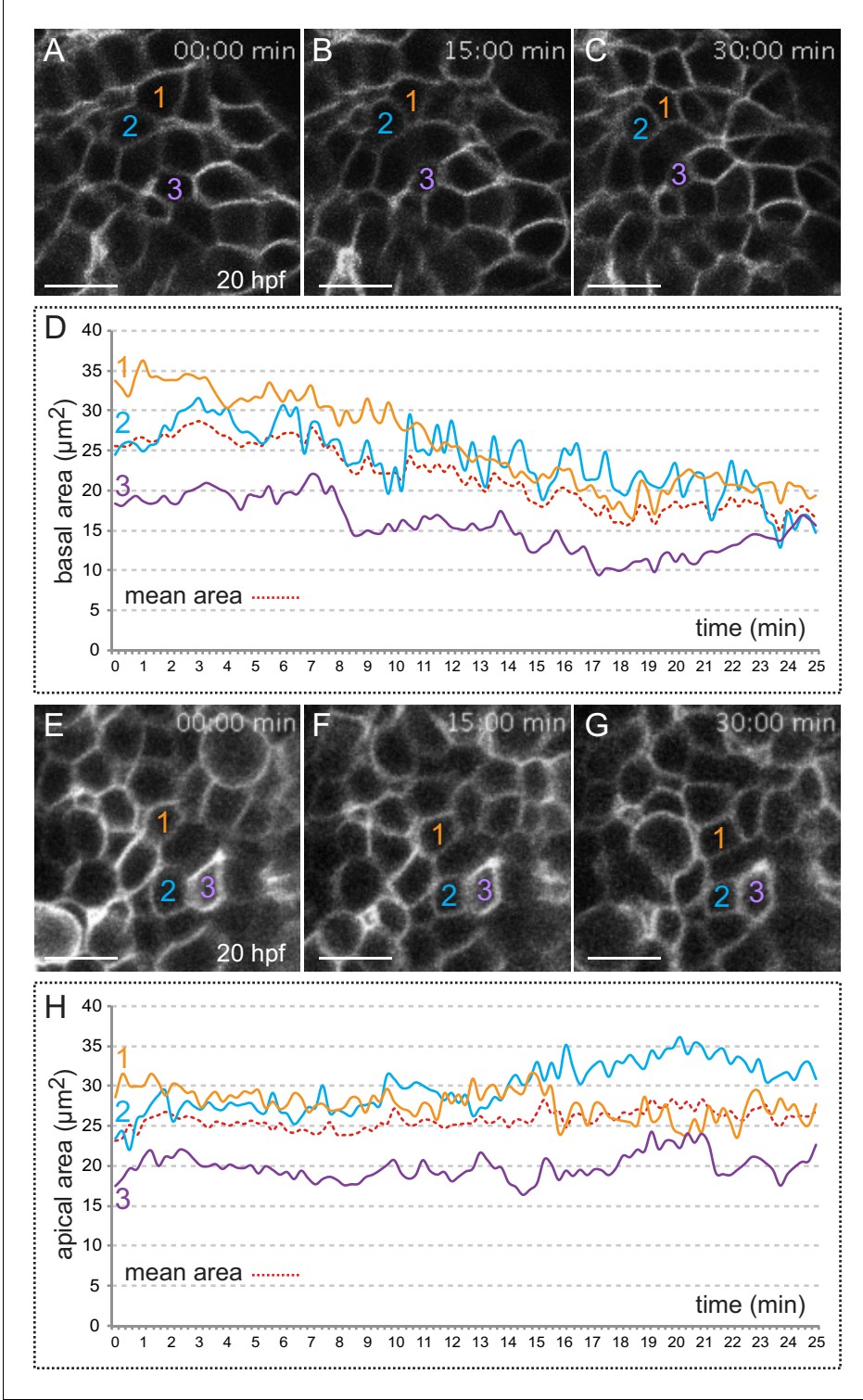

**Figure 2.** Quantitative analysis of membrane oscillations in tg(vsx2.2:GFP-caax) embryos. Cell area dynamics at the basal (**A**–**D**) and apical (**E**–**H**) surfaces is shown for three individual cells (color coded). Absolute basal (**D**) and apical (**H**) areas in μm$^2$ are represented versus time for the individual cells. The mean area indicates a progressive constriction of the basal, but not apical surfaces over time (**D**, **H**). Scale bars = 10 μm.

The following figure supplement is available for figure 2:

**Figure supplement 1.** Quantitative analysis of cell pulses.

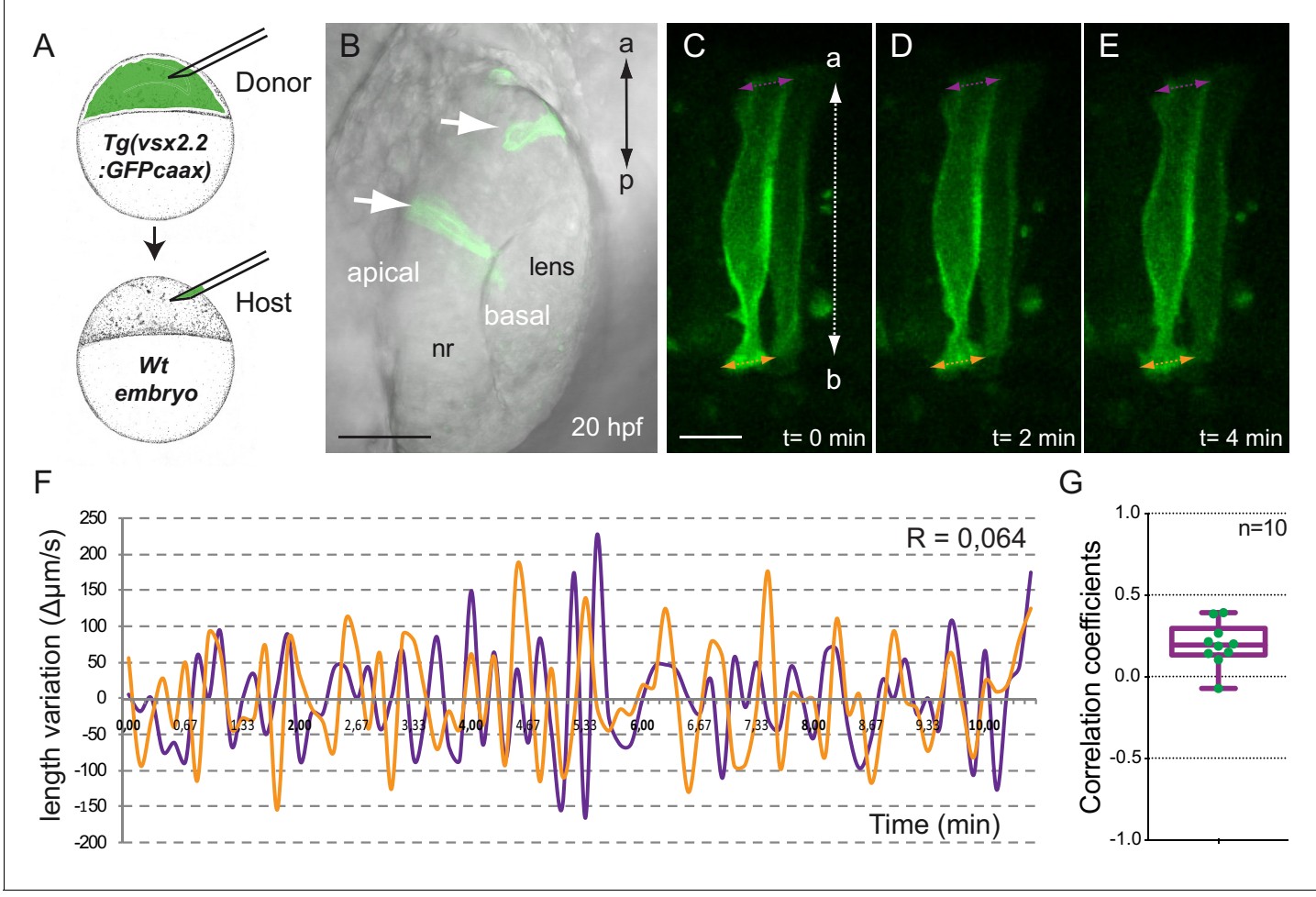

**Figure 3.** Analysis of tg(vsx2.2:GFP-caax) clones show uncoupled oscillations at apical and basal surfaces. (**A**) Scheme of transplantation experiment at sphere stage. (**B**) Confocal microscopy image showing transmitted light and GFP expression for transplanted clones (white arrows) at 20 hpf. Antero-posterior (**a**–**p**) axis is indicated. (**C**–**E**) Confocal microscopy time-lapse images show length variation of basal (orange) and apical (purple) edges through time in a transplanted clone. The orientation of the apico-basal (**a**–**b**) axis is indicated. Scale bars = 50 μm in B and 10 μm in C–E. (**F**) Quantification of the basal (orange) and apical (purple) length variation for an individual clone showing no correlation between the oscillations (R = 0064). (**G**) Box plot showing the distribution of apical vs basal oscillations correlation coefficients for 10 transplanted neuroblasts from five different retinas.

The following figure supplement is available for figure 3:

**Figure supplement 1.** Mitotic rounding impact on basal constriction and apical expansion.

observation confirms the basal constriction of the retinal neuroepithelium during optic cup morphogenesis.

## Apical and basal surfaces behave as independent oscillators and mitoses result only in a transient expansion of the apical domain

As periodic contractions occur at both neuroblasts' ends with a similar frequency, we next ask whether apical and basal surfaces oscillate synchronically. To answer this issue, we generated retinal clones by blastomere transplantation from *tg(vsx2.2:GFP-caax)* donor embryos into wild-type late-blastula hosts. Live-imaging analysis of singularized *tg(vsx2.2:GFP-caax)* neuroblasts along the apico-basal axis allowed the simultaneous recording of variations in the length of the apical and basal edges at 20 hpf (*Video 3*). Quantitative analysis of 10 individual cells revealed a poor correlation between the pulses at apical and basal ends (R < |0.5| in all cells examined), thus indicating that

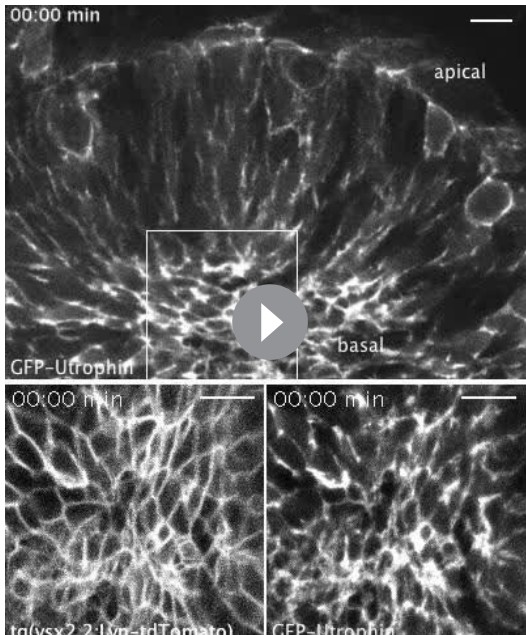

**Video 4.** Actin dynamics in constricting retinal cells. (Upper panel) Maximum projection of 3 z-stacks (over a total of 1 µm) along the apico-basal axis shows actin oscillatory activity in *tg(vsx2.2:lyn-tdTomato)* embryos at 20 hpf. Retinal basal surface (region within the square) is magnified in lower panels. (Lower panels) Time lapse shows the simultaneous recording of membrane behavior, as revealed by *lyn-tdTomato* (left panel), and actin dynamics, as revealed by Utrophin-GFP (right panel). Images were acquired every 5 s. Scale bars = 10 µm. See also *Figure 4*.

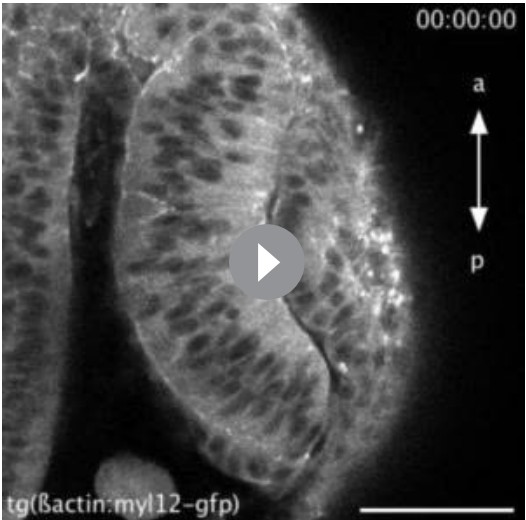

**Video 5.** Myosin dynamics during optic cup morphogenesis. Live imaging analysis of *tg(actb1: myl12.1-eGFP)* embryos reveal myosin accumulation at apical lens and basal retina epithelia. Movie starts at 19 hpf. Antero-posterior (a-p) axis is indicated. Images were acquired every 20 s. Scale bar 50 µm. See also *Figure 5*

these surfaces oscillate largely in an independent manner (*Figure 3*). A second emerging question was whether apical cell rounding during mitosis may affect either basal constriction or apical expansion. To address this issue, the distance between the two cells flanking mitotically active neuroblasts was measured through time. Whereas quantitative analysis of distance variation showed a transient expansion of the apical domain as the cells divide, this was recovered once mitoses were resolved (*Figure 3—figure supplement 1*). Thus, both the apical and basal net distances at the beginning and end of the process did not change significantly (T-test; n = 10). This observation is in agreement with previous data showing that cell mitoses did not play a major role for optic cup formation (*Kwan et al., 2012*).

## Actin dynamics in constricting retinal cells

Oscillatory cell contractions and epithelial bending have been associated to the periodic accumulation of the cortical actomyosin network. To investigate this phenomenon in constricting retinal cells, we first examined actin dynamics during optic cup morphogenesis. To follow dynamic changes in cell area and F-actin simultaneously, we injected *utrophin-GFP* RNA in one-cell stage embryos of the transgenic line *tg(vsx2.2:lyn-tdTomato)* and then performed live imaging analyses at 20 hpf focusing on the basal neuroblasts surface (*Video 4*). As previously reported for vertebrate neuroepithelial cells (*Nishimura et al., 2012*), actin accumulated circumferentially (i.e. junctional) rather than medially as observed in constricting *Drosophila* epithelia (*He et al., 2010; Martin et al., 2009*) (*Figure 4A–F*). In addition, we observed that actin accumulated at the basal surface and oscillated with a frequency similar to membrane pulses (*Video 4*). To detect whether there is a relationship between cortical actin accumulation and basal area changes, both parameters were quantified after segmentation and a cross-correlation analysis was performed. This analysis showed a positive association between actin accumulation and basal area expansion, with a cross-correlation coefficient of 0.40 ± 0.16 (median 0.35), as calculated for 26 cells from three different experiments (*Figure 4G,H*).

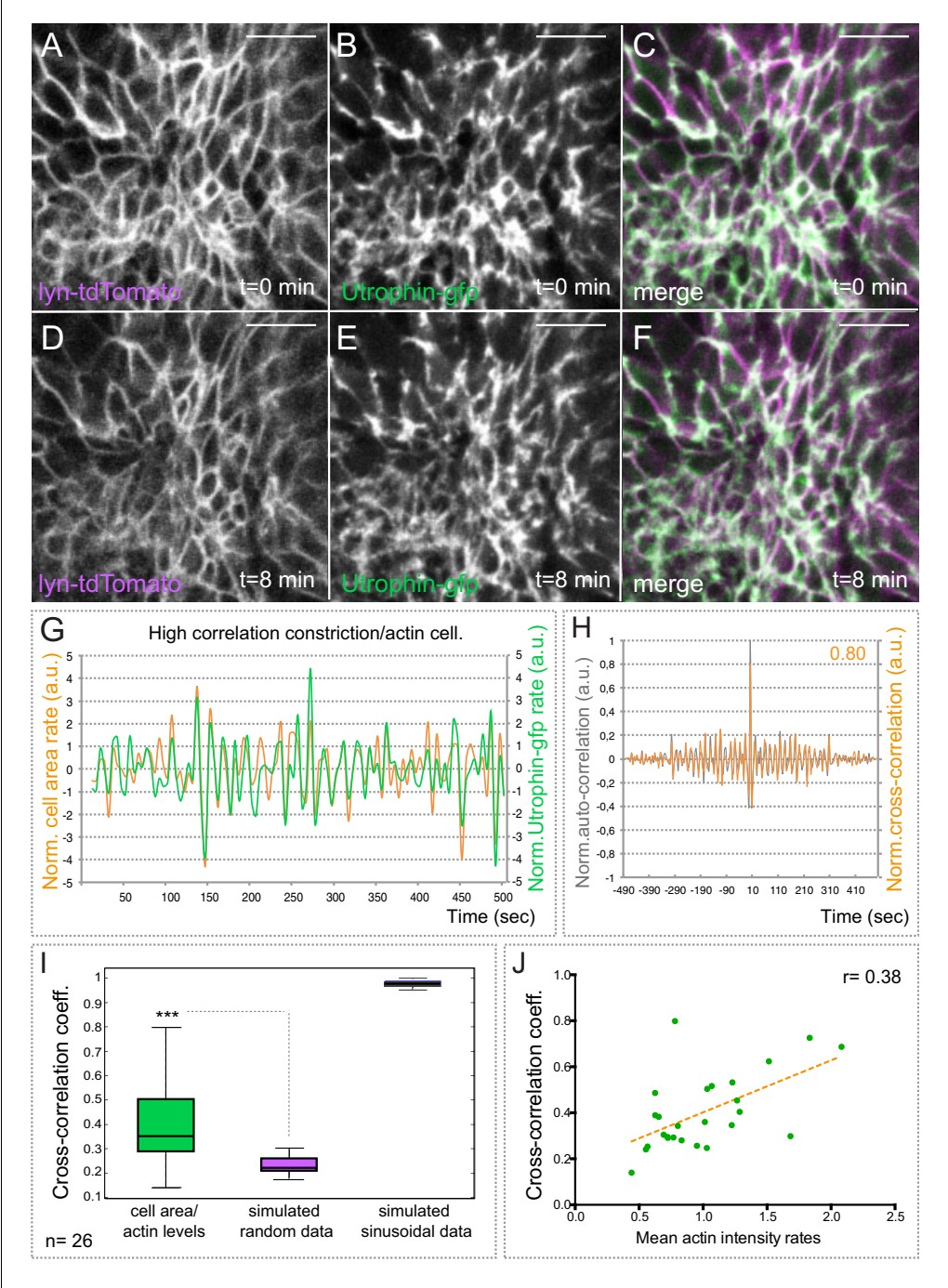

**Figure 4.** Basal actin dynamics in constricting retinal cells. (A–F) Actin dynamics, as revealed by utrophin-gfp, and membrane oscillations were simultaneously examined by time lapse in the line tg(vsx2.2:lyn-tdTomato) at 20 hpf (see **Video 4**). Note that F-actin localizes mainly at the cellular cortex. Scale bars = 10 μm. (G) Normalized basal area rate (orange) and normalized utrophin-gfp rate (green) are shown over time for a cell displaying a high correlation between actin oscillations and membrane expansion. Area rate and Utrophin-gfp rate were normalized dividing by the mean of their absolute values. (H) Normalized auto-correlation (grey line) and cross-correlation (orange) are shown for cell represented in G. Maximum cross-correlation (0.8) is indicated. (I) Box plot comparison of cross-correlation results between actin vs. membrane oscillations, simulated random and simulated sinusoidal signals shows a significant (p<0.001; T-test; n = 26) positive correlation between actin accumulation and basal area expansion. (J) Scattered plot showing the dependency of cross-correlation coefficients (n = 26) on mean actin intensity rates. Linear regression line (orange) and linear correlation coefficient (0.38) are indicated.

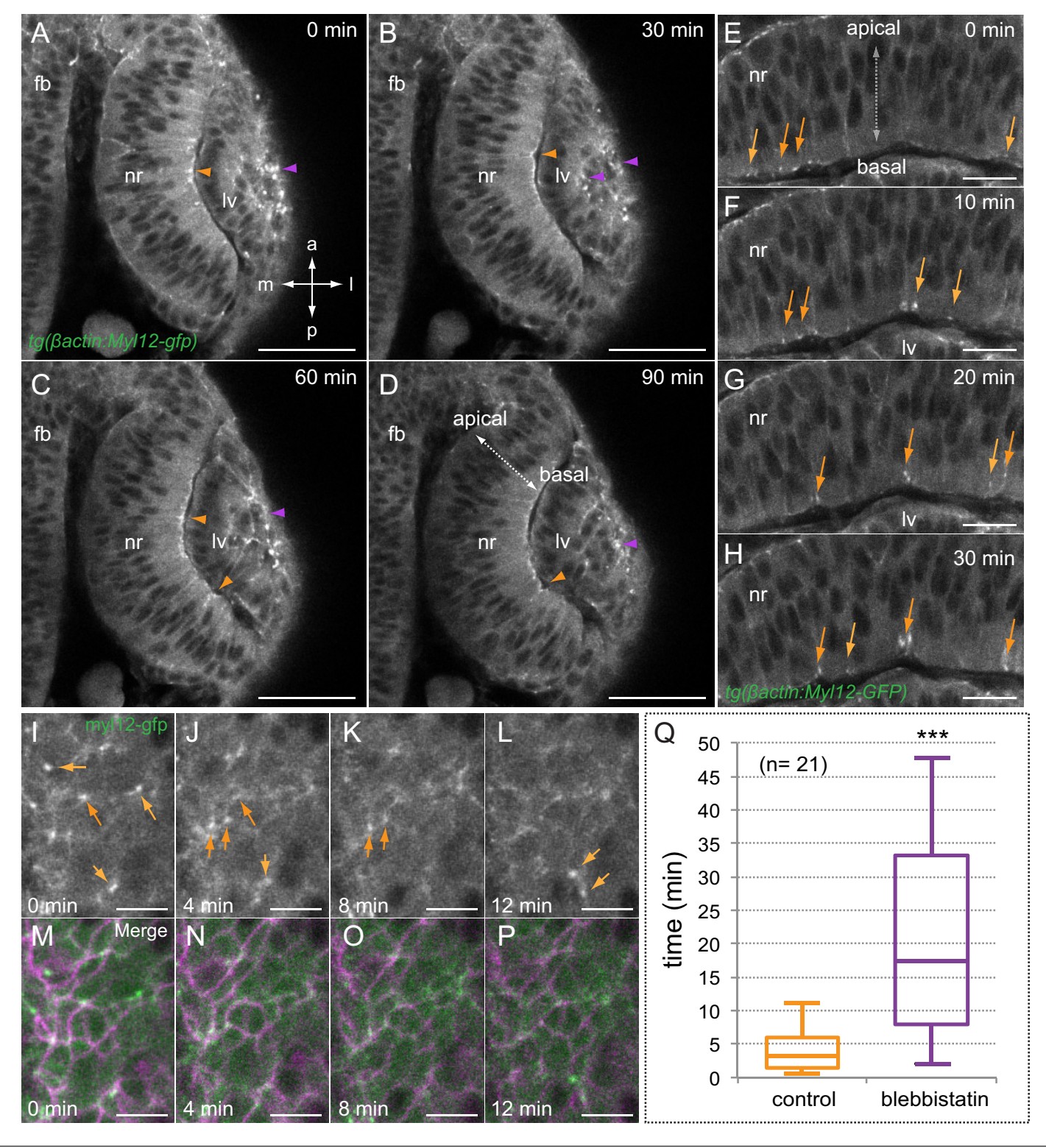

**Figure 5.** Myosin accumulates in basal foci during optic cup morphogenesis. (A–D) Live-imaging analysis of tg(actb1:myl12.1-eGFP) embryos reveals myosin accumulation at the apical lens (purple arrowheads) and basal retina (orange arrowheads) between 19 and 20.5 hpf. Antero-posterior (a–p) and medio-lateral (m-l) axes are indicated. (E–H) Myosin accumulates in transient foci (orange arrows) at the basal cortex. (I–P) Time-lapse analysis of myosin foci at the basal surface plane in embryos injected with lyn-tdTomato RNA reveals that the protein accumulates at the peripheral cortex in scattered

*Figure 5 continued on next page*

*Figure 5 continued*

cells. (**Q**) The box plot shows a significant difference in foci stability between control and blebbistatin (150 µM) treated embryos (T-test, n = 21). fb = forebrain; nr = neural retina; lv = lens vesicle. Scale bars = 50 µm in **A–D**, 20 µm in **E–H**, and 10 µm in **I–P**.

In order to evaluate the significance of our results, we compared our experimental data with simulated random and sinusoidal signals of similar statistical properties. Coefficients of simulated random data were significantly lower than our observations in vivo, indicating that the cells display a significant positive correlation between actin accumulation and basal area changes (*Figure 4I*). Hence, cell area expansion and actin accumulation occur simultaneously or with time lags shorter than 5 s (i.e. our sampling rate limitation). Furthermore, when we plotted cross-correlation coefficients as a function of the actin intensity, we observed higher coefficients corresponding to cells with higher actin intensity rates (*Figure 4J*). Taken together, these results indicate that the molecular mechanism responsible for the fast oscillations in the vertebrate retina differs in important aspects from that controlling the pulsatile behavior in constricting epithelia in *Drosophila*. In retinal neuroblasts, peripheral actin accumulation is associated with basal ends' expansion, whereas in *Drosophila* cell contraction is linked to medial condensation of actin.

## Myosin dynamics in constricting retinal cells

To investigate myosin dynamics, we then carried out time-lapse studies through optic cup folding in *tg(actb1:myl12.1-eGFP)* embryos. At the organ level, myosin accumulations were detected both at the apical lens and basal retina epithelia (*Figure 5A–H*; *Video 5*). This is in agreement with the bending of these tissues toward their apical and basal surfaces, respectively. When examined in relation to basal membrane oscillations, as revealed by *lyn-tdTomato*, myosin dynamics showed a behavior different from that of actin. Basal myosin accumulates in scattered cortical foci, which have an average stability in the range of minutes, 4 ± 0.5 min (*Figure 5I–Q*). Treatment of embryos for 1 hr with blebbistatin, a specific inhibitor that blocks myosin in an actin-detached state (*Kovacs et al., 2004*), severely interfered

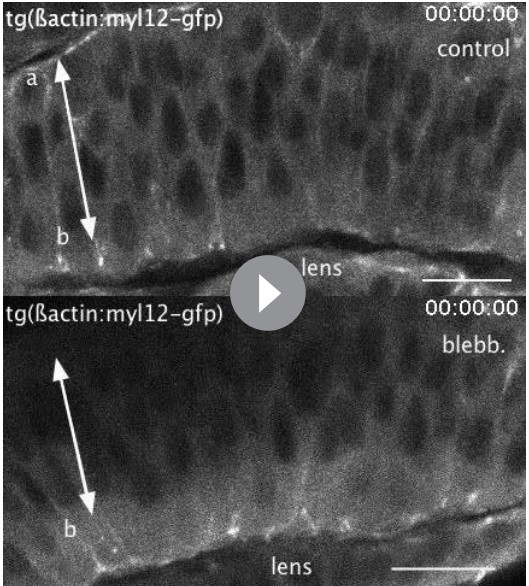

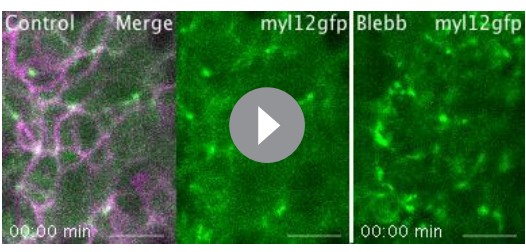

**Video 6.** Myosin foci dynamics at the basal surface. Live-imaging analysis of myosin distribution at the basal surface in 20 hpf *tg(actb1:myl12.1-eGFP)* embryos shows cortical localization of myosin foci in scattered cells (left and middle panels). Membrane oscillations were simultaneously examined by injection of *lyn-tdTomato* RNA (merged in left panel with myl12gfp). Treatment of *tg(actb1:myl12.1-eGFP)* embryos with blebbistatin (150 µM) severely blocks myosin dynamics at the basal surface (right panel). Images were acquired every 5 s. Scale bar 10 µm. See also *Figure 5*.

**Video 7.** Myosin foci dynamics and basal membrane indentations upon blebbistatin treatment. Live-imaging analysis of myosin dynamics at the basal surface both in control (upper panel) and blebbistatin treated (150 µM; lower panel) 20 hpf embryos from the line *tg(actb1: myl12.1-eGFP)*. Note the increased stability of the myosin foci and the reduced contractility of the basal surface in the retina of the blebbistatin-treated embryos. Images were acquired every 10 s Scale bar = 10 µm. See also *Figure 6*.

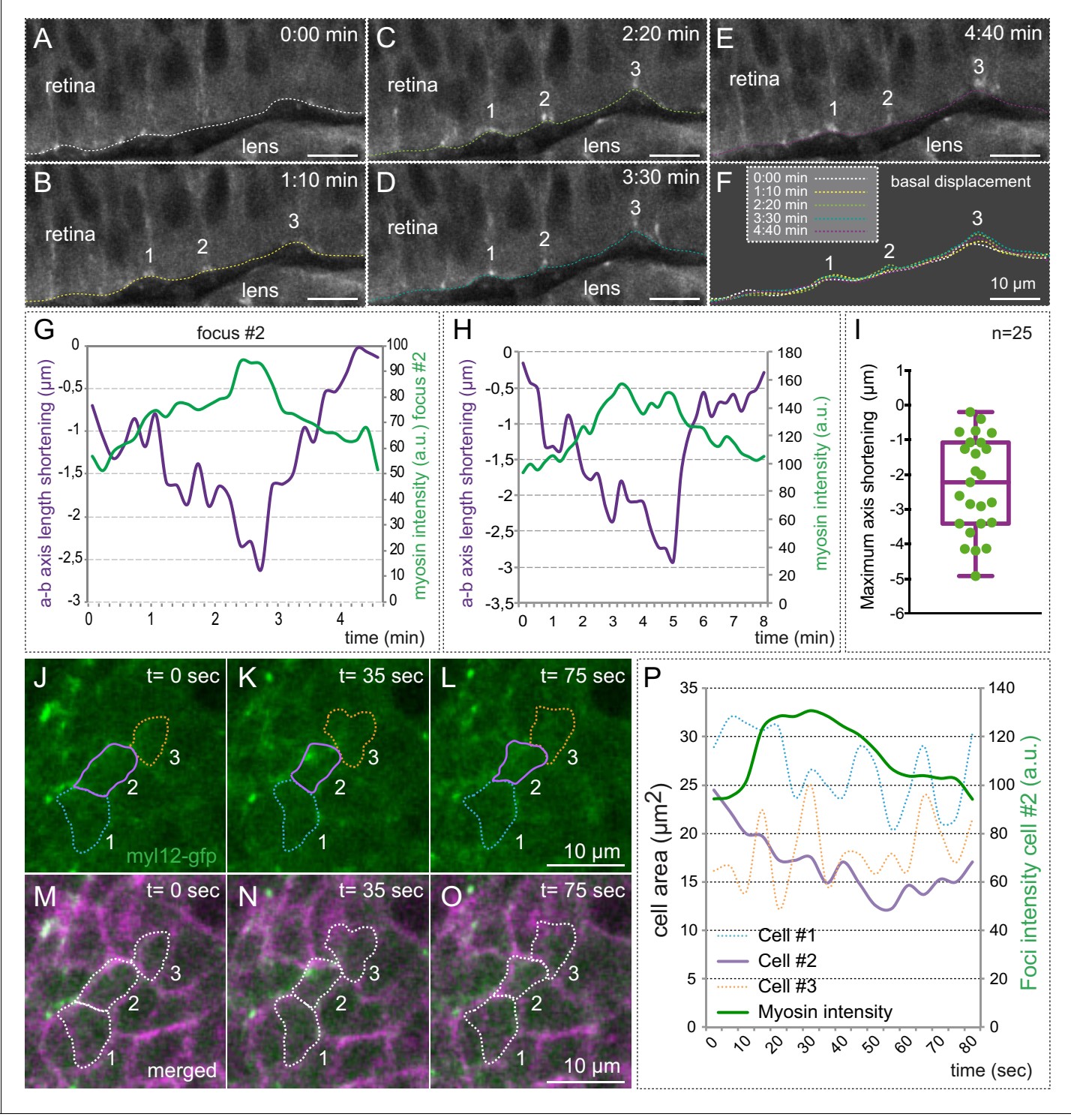

**Figure 6.** Myosin accumulation correlates with basal membrane displacement. (**A**–**E**) Time series of optical sections from tg(actb1:myl12.1-eGFP) embryos show discrete myosin foci (labeled 1, 2, 3) and basal surface displacement. (**F**) Basal edges were color-coded for each time point and overlapped to illustrate the transient indentations of the basal surface associated to myosin foci. (**G**–**H**) Quantitative recording over time of myosin intensity and apico-basal axis shortening for a couple of representative foci. The focus in **G** is #2 in **A**–**F**. (**I**) Box plot showing the maximum shortening of the a-b axis for 25 foci from 12 different retinas. (**J**–**O**) Correlative analysis of basal area (revealed by lyn-tdTomato) and myosin dynamics is shown for three neighbor cells (color-coded). (**P**) Quantitative analysis of cell area changes and myosin intensity for the three neighboring cells. Note that only the cell accumulating myosin contracts. Scale bars =10 μm.

*Figure 6 continued on next page*

*Figure 6 continued*

The following figure supplements are available for figure 6:

**Figure supplement 1.** Myosin accumulation correlates with basal contraction.

**Figure supplement 2.** Myosin inhibition impairs basal constriction.

**Figure supplement 3.** Myosin inhibition interferes with optic cup folding.

with myosin dynamics in the retina, increasing significantly the stability of the foci to 21.5 ± 2.4 min (*Figure 5Q*, *Video 6*).

Live-imaging analysis along the apico-basal retinal axis showed that myosin foci correlate with basal membrane indentations (i.e. transient shortenings of the apico-basal axis), suggesting active pulling of the basal lamina (*Figure 6A–F*; *Video 7*). To quantitatively analyze this phenomenon, we measured simultaneously myosin intensity and apico-basal axis shortening (*Figure 6G–H*). The analysis of 25 individual foci revealed a significant shortening of the apico-basal axis upon myosin accumulation for most of the events examined, with an average shortening of 2.3 ± 1.4 (SD) µm (*Figure 6I*). Correlative analysis of basal membrane dynamics and myosin accumulation in *tg(actb1:myl12.1-eGFP)* embryos injected with *lyn-tdTomato* RNA revealed that a large proportion of the cells containing myosin foci contract significantly their basal surface (*Figure 6J–P*; *Figure 6—figure supplement 1*). In contrast, the oscillatory behavior and average area of the cells neighboring those with myosin foci was not affected upon myosin accumulation (*Figure 6P*; *Figure 6—figure supplement 1*).

As we mentioned, myosin inhibition stabilized cortical myosin foci. Blebbistatin treatment also impaired contractility at the basal surface of the retina. Thus, basal membrane indentations associated to myosin foci appeared largely

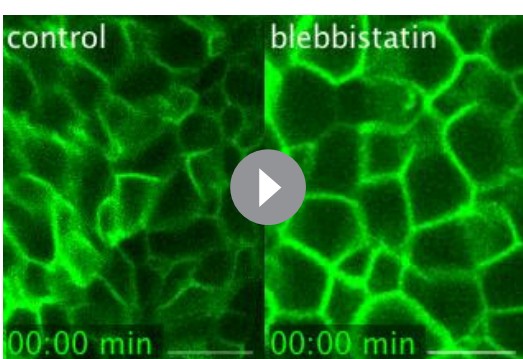

**Video 8.** Membrane oscillations at the basal surface in control and blebbistatin-treated embryos. Maximum projection of 3 z-stacks (over a total of 1 µm) at the basal surface in *tg(vsx2.2:GFP-caax)* retinae show cell membranes oscillatory behavior over a period of 25 min in control (left panel) and blebbistatin treated (150 µM; right panel) embryos. Note that blebbistatin treatment abolishes the oscillatory behavior and blocks the cells in a relaxed state. Images were acquired every 5 s. Scale bars = 10 µm. See also *Figure 6—figure supplement 1*.

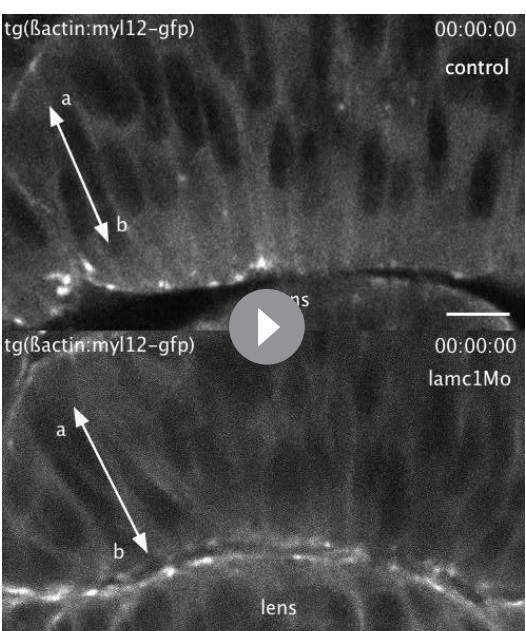

**Video 9.** Myosin foci dynamics and basal membrane indentations in wild-type and *lamc1* morphants. Live-imaging analysis of myosin dynamics at the basal surface both in control (upper panel) and *lamc1Mo*-injected (lower panel) 20 hpf embryos from the line *tg(actb1:myl12.1-eGFP)*. Note the increased stability of the myosin foci and the reduced contractility of the basal surface in *lamc1* morphants. Images were acquired every 10 s Scale bar = 10 µm. See also *Figure 7*.

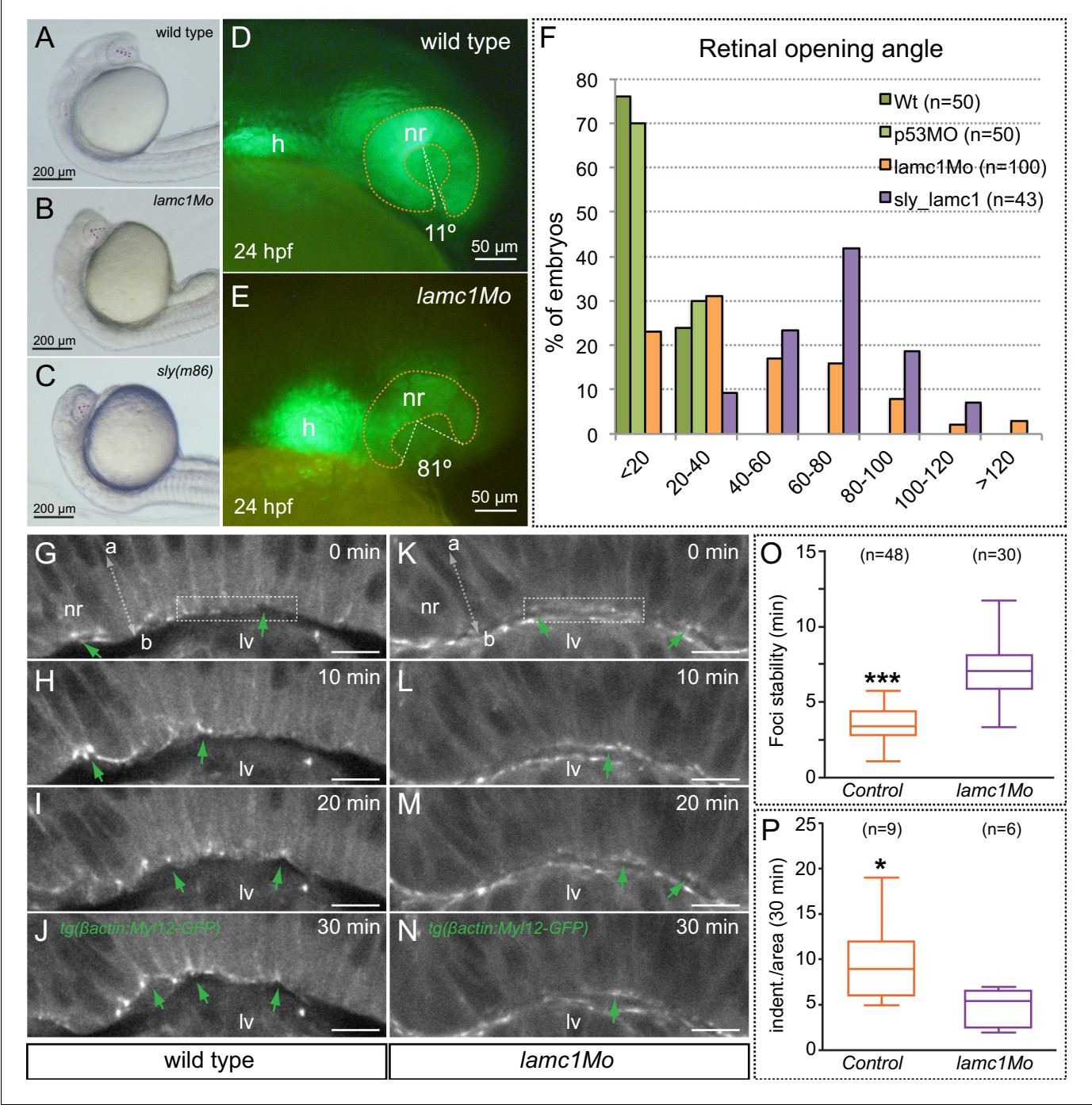

**Figure 7.** Optic cup folding, basal contractility and myosin dynamics depend on lamc1 function. (A–C) General embryo morphology for wild type, lamc1 morphants and sly (lamc1-/-) mutants at 24 hpf. Retinal opening is indicated with a dashed line. (D–E) Retinal morphology in tg(vsx2.2:GFP-caax) both wild type and lamc1Mo-injected embryos, at 24 hpf. Ventral opening angle (white) and retinal contour (orange) are indicated with dashed lines. (F) Frequency distribution of retinal opening angles is shown for controls (either wild type or p53Mo-injected), lamc1Mo injected, or sly mutants. (G–N) Time-lapse analysis of tg(actb1:myl12.1-eGFP) wild type and lamc1Mo-injected embryos show dynamic accumulation of myosin foci (green arrows) at the basal surface. (O) Analysis of myosin foci reveals that they are significantly more stable in lamc1Mo-injected embryos (T-test). (P) The box plot shows that transient indentations of the basal surface are significantly diminished in lamc1Mo-injected embryos (T-test). h = heart; nr = neural retina; lv = lens vesicle. Scale bars = 200 µm in A–C, 50 µm in D–E, and 10 µm in G–N.

The following figure supplement is available for figure 7:

**Figure supplement 1.** Analysis of membrane oscillations reveal impaired basal constriction in lamc1 morphant embryos.

attenuated (*Video 7*), suggesting an inefficient mechanical coupling. In addition, when basal membrane oscillations were examined in the *tg(vsx2.2:GFP-caax)* line, treatment for one hour with blebbistatin abolished the pulsatile behavior and impaired basal constriction by blocking the cells in a relaxed state (*Figure 6—figure supplement 2*, *Video 8*). This result indicates that although myosin levels do not oscillate with basal area changes, its activity is required to maintain the pulsatile dynamics. Finally, sustained treatment with blebbistatin for 3 hr significantly delays the folding of the optic cup (*Figure 6—figure supplement 3*). This finding, however, needs to be interpreted cautiously, as myosin inhibition may interfere with optic cup folding either by blocking basal constriction or through any other acto-myosin-dependent morphogenetic mechanism.

## Lamc1 function is required for efficient cell contractility, basal constriction and optic cup folding

We have previously shown that integrin-mediated adhesion to the ECM plays a fundamental role during optic cup folding in medaka (*Martinez-morales et al., 2009*; *Bogdanovic et al., 2012*). To specifically interfere with this process in zebrafish, we knocked down *lamc1*, a core component of laminin trimer, the mutation of which results in ocular malformations (*Domogatskaya et al., 2012*; *Lee and Gross, 2007*). To this end we employed morpholinos previously reported to phenocopy the zebrafish *lamc1* mutation *sleepy* (*sly*) (*Parsons et al., 2002*; *Ivanovitch et al., 2013*). Comparative examination of *sly* mutants and *lamc1* morphants revealed a similar optic cup phenotype (*Figure 7A–C*), both interfering with the folding of the epithelium, as indicated by measurement of retinal opening angles at 24 hpf (*Figure 7D–F*). Live-imaging analysis of *tg(vsx2.2:GFP-caax)* morphant retinas revealed that basal oscillations are not reduced upon *lamc1* knockdown; on the contrary, their average peak amplitude was significantly increased by 45% (n = 22). Interestingly, the progressive reduction of the cellular feet observed in control retinas (*Figure 2*) was severally impaired in embryos injected with *lamc1* morpholinos (*lamc1Mo*), and basal cell areas appeared significantly larger when compared to the control situation (*Figure 7—figure supplement 1*). This observation indicates that laminin-dependent adhesion to the ECM is required for effective basal constriction.

To investigate myosin dynamics in the folding retina of *lamc1*-deficient embryos, morpholinos were injected in the *tg(actb1:myl12.1-eGFP)* line. Live-imaging analysis of *lamc1Mo* and control sibling embryos revealed that myosin foci are still observed in the morphant retinae (*Figure 7G–N*). However, in the morphant tissue, foci were significantly more stable than in the wild-type siblings, and more importantly, basal membrane indentations associated to them appeared attenuated (*Figure 7O,P*; *Video 9*). This result suggests that deficient adhesion to the ECM also results in a less efficient transmission of mechanical tensions and hence reduced contractility at the basal feet.

## Analysis of tension distribution during optic cup morphogenesis by laser ablation

To examine how mechanical tensions are distributed in the folding epithelium, we performed laser ablations experiments at different stages of

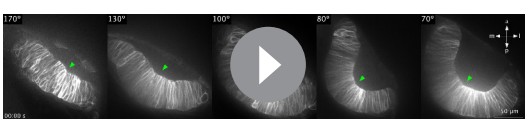

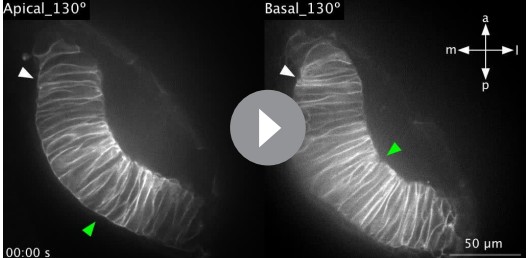

**Video 10.** Laser ablation experiments at the basal surface of the retina through optic cup folding. Local cell ablations were carried out in tg(vsx2.2:GFP-caax) retinae at different stages. Ablation points are indicated with green arrowheads. Retinal folding angles are indicated. Note the global tissue relaxation upon ablation at 130°. Images were acquired every seconds. Scale bar = 50 µm.

**Video 11.** Comparative analysis of focal ablations at the apical or basal surface of the retina. Ablations were carried out in tg(vsx2.2:GFP-caax) retinas with a 130° opening. Ablation points are indicated with green arrowheads. Peripheral tissue displacement is indicated with white arrowheads. Scale bar = 50 µm.

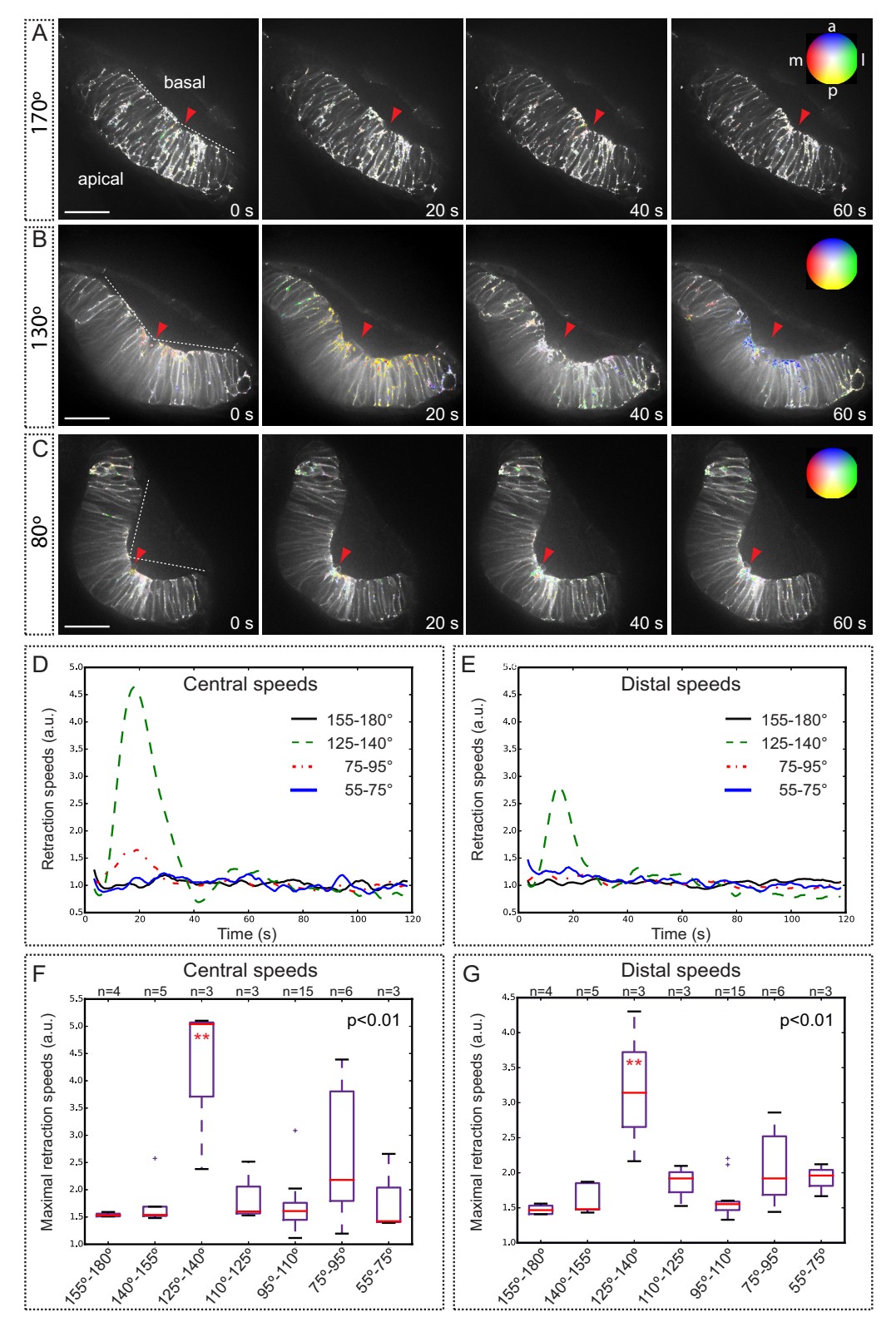

**Figure 8.** Optical flow analysis of tissue displacement upon laser ablation at different stages of folding. (A–C) Analysis of pixel displacement after laser ablation at the basal surface is shown for retinas at 170°, 130°, and 80° of bending. Red arrowheads indicate the ablation point. Particles' motion vectors are indicated with a color code: Colors correspond to the direction of the displacement and color intensity to its magnitude. Note maximum displacement 20 s after ablation in 130°-stage retina. Scale bar = 50 µm. See *Video 12*. (D–E) Average tissue retraction speed profiles over time are

*Figure 8 continued on next page*

*Figure 8 continued*

shown for different stages of optic cup folding (represented as angle bins), both at the central (D) or distal (E) positions in the retina. (F–G) Box plot representation of maximal retraction speeds at the different stages, represented as angle bins. For each stage, median values (red bars) and sample sizes are indicated. One-way ANOVA analysis followed by Dunnett's multiple comparison tests show significant differences ($p < 0.01$**) only at 125–140°-stage.

The following figure supplements are available for figure 8:

**Figure supplement 1.** Tissue local relaxation upon laser ablation: Optical flow analysis of tissue displacement.

**Figure supplement 2.** Optical flow analysis of retinal tissue displacement upon apical vs basal laser ablation.

**Figure supplement 3.** Optical flow analysis of tissue displacement upon laser ablation in wild type vs. lamc1_Mo tissues.

optic cup morphogenesis. In order to visualize membranes displacement during tension release, local ablations were carried out in *tg(vsx2.2:GFP-caax)* embryos, either at the apical or at the basal surfaces of the tissue. Laser-induced cuts trigger a limited expansion of the wounded area and a local relaxation of the tissue, as determined by optical flow analysis (*Figure 8—figure supplement 1*). For most of the stages analyzed, tissue relaxation affected only neuroblasts immediately adjacent to the wounded area. However, laser ablations within a developmental window corresponding to a 125°–140° opening of the optic cup resulted in a global tissue relaxation that affected bending of the entire epithelium (*Video 10*). At this specific stage, tension release triggered a noticeable folding of the retinal tissue toward its basal surface. To quantitatively investigate membrane displacement after laser ablation in retinal tissues, we carried out an optical flow analysis of the movies (*Figure 8A–C*; *Video 12*), which allow determining retraction speeds at different stages and locations within the tissue (*Figure 8D,E*; *Figure 8—figure supplement 1*). Statistical analysis of optical flow data confirmed that maximum retraction speeds are significantly higher only for retinas displaying a 125°–140° bending (*Figure 8F,G*). This observation indicates that the balance between tensile forces and tissue resistance that maintains organ shape is particularly unstable within a narrow developmental window that coincides with the acute constriction of the basal feet at 19 hpf (*Figure 1*). In contrast to the global reaction observed upon basal ablation, which triggers the displacement of the peripheral retina, apical ablation only affected the morphology of the central retina but no peripheral retraction was observed (Movie 11; *Figure 8—figure supplement 2*). The differential tissue response upon ablation at the apical and basal surfaces, together with our previous observations on *lamc1* requirement for basal contractility (*Figure 7*) prompted us to investigate tissue behavior in *lamc1* morphants. The analysis of retraction speeds in laser-ablated tissues at the critical 125°–140° stage showed that global relaxation of the optic cup is attenuated in *lamc1* knockdown retinas (*Figure 8—figure supplement 3*). This data

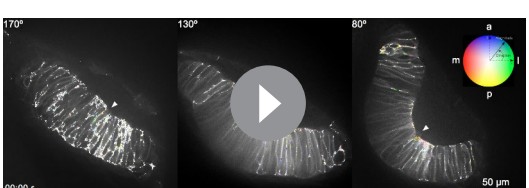

**Video 12.** Optical flow analysis of tissue displacement upon laser ablation at different stages of optic cup folding. Ablation points are indicated with white arrowheads. Particles' motion vectors are indicated with a color code. Images were acquired every seconds. Scale bar = 50 μm.

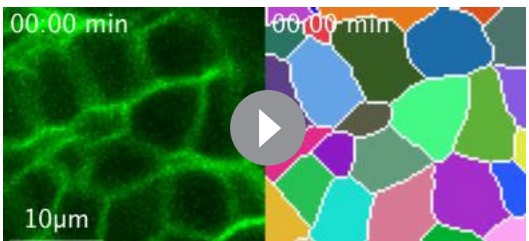

**Video 13.** Membrane oscillations in an optical section from a *tg(vsx2.2:GFP-caax)* embryo and Packing Analyzer v2.0 automatic cell edge detection (represented by unique RGB codes) are shown in parallel movies. Scale bar = 10 μm. See also *Figure 1_figure supplement 1*.

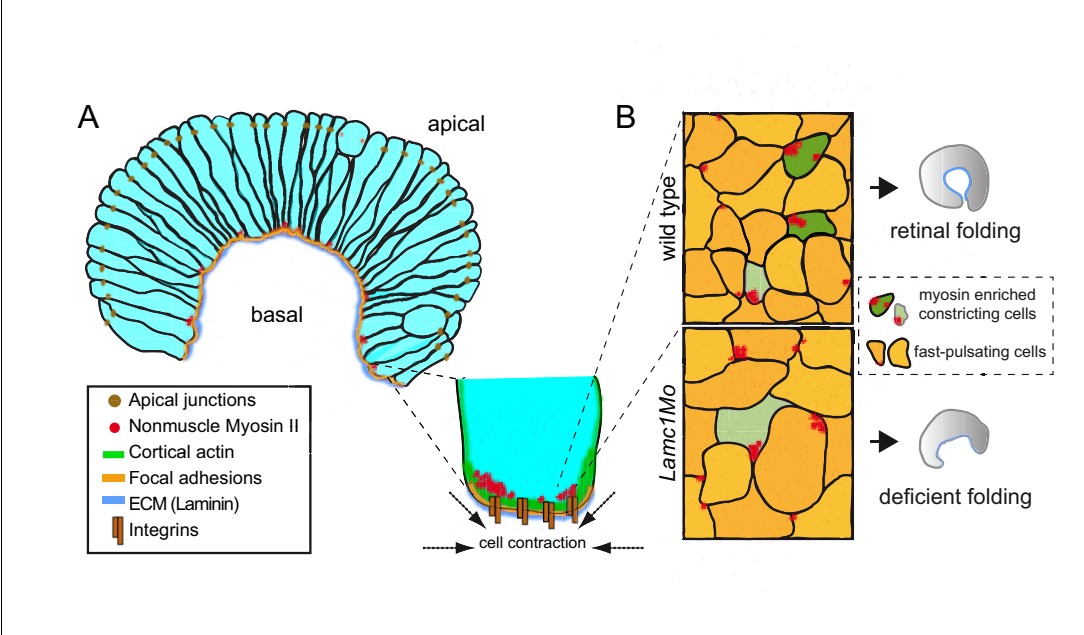

**Figure 9.** A working model for the basal constriction of the retinal epithelium. (**A**) Representation of the retinal epithelium during eye morphogenesis showing the distribution of cortical actomyosin, integrins and ECM at the basal surface of the tissue. Apical junctions and focal adhesion components have been included as a reference for apico-basal polarity. (**B**) Schematic diagram representing the condensation of nonmuscle myosin II foci at the basal surface in wild type and *lamc1Mo* retinas. Both fast pulsating cells (orange) and myosin-enriched constricted cells (green) are depicted. Weakly constricting neuroblast feet are represented in pale green. The final form of the organ is also shown for wild type and lamc1 deficient embryos.

indicates that the laminin-mediated attachment to the ECM is essential for the transmission of mechanical tensions throughout the folding tissue.

## Discussion

In the current study, we have characterized the morphogenetic behavior of retinal precursors during zebrafish optic cup folding by live imaging. Our quantitative analysis demonstrates that retinal neuroblasts undergo a progressive constriction of their basal surface. Previous reports have described the involution of outer layer progenitors into the presumptive neural retina domain as a mechanism driving the formation of the eye chamber (*Picker et al., 2009; Kwan et al., 2012; Heermann et al., 2015*). Our observations are also consistent with these reports, thus suggesting that basal constriction and cell involution cooperate during eye morphogenesis in zebrafish. Comparative analysis of our data and previous studies (*Heermann et al., 2015*) indicate that, although both mechanisms overlap substantially, they are staggered events. Whereas basal constriction occurs mainly during the primary folding of the retinal epithelium between 18 and 20 hpf, cell involution through the rim is limited during this period and becomes more prominent at later stages between 20 and 24 hpf. It is tempting to speculate that these mechanisms might be coupled. Thus, basal constriction may generate centripetal tensions facilitating cell involution and, conversely, cell involution may relieve tissue resistance supporting a constriction-dependent optic cup folding. However, the precise cellular mechanisms driving cell involution are currently unknown, and hence exploring this possibility will require further investigation.

Here, we have described that retinal precursors undergo fast pulsations both at their apical and basal surfaces. We then examined both membrane and actomyosin dynamics at the basal surface, where the progressive constriction takes place. Although, in principle, the neuroblasts' periodic pulsations share some features with the oscillations observed in other constricting epithelia (*Kim and Davidson, 2011; Martin et al., 2009; Roh-johnson et al., 2012; Solon et al., 2009*), there are fundamental differences. In most epithelial cells, pulsations are more regular in frequency and

amplitude than in retinal precursors, and their average oscillation frequency range between 1 and 5 min (*Gorfinkiel and Blanchard, 2011*). This is in contrast to irregular fast oscillations (≈20 s) here described in the zebrafish retina. A second fundamental difference concerns the organization of the actomyosin fibers in the shrinking surface of the tissue. In most of the constricting epithelia so far examined, contractile actomyosin fibers accumulate in a medioapical domain. From this domain, centripetal tension responsible for cell contraction is generated and transmitted to surface junctions (*Martin et al., 2009; Roh-johnson et al., 2012; He et al., 2010*). Interestingly, F-actin turnover is required for this medioapical localization of the actomyosin meshwork, its efficient attachment to cellular junctions, and the generation of centripetal tension (*Jodoin et al., 2015*). In contrast, our data show that both actin and myosin fibers accumulate at the cellular cortex in the zebrafish retina. Cortical distribution of actomyosin fibers has also been described in the folding neural tube (*Nishimura et al., 2012*), thus suggesting that it may be a common feature in elongated neuroepithelial cells regardless the tissue is bending toward its apical o basal surface.

It has been shown that medial and cortical actomyosin pools have different mechanical properties in epithelial cells (*Rauzi et al., 2010*). In the light of this finding, our observation that the molecular mechanism driving fast oscillations in retinal neuroblasts differs substantially from that previously reported in constricting epithelia is not surprising. Whereas medioapical accumulations of actomyosin precede periodic cellular contractions in most epithelia analyzed, we observed that cortical actin accumulation correlates positively with basal membrane expansion in retinal precursors. Local actin assembly at the leading edge has been described as a positive force driving membrane extension in lamellipodia and axonal growth cones (*Pollard and Borisy, 2003; Levayer and Lecuit, 2012; Medeiros et al., 2006*). Our data may suggest a similar mechanism as responsible for the pulsatile behavior of the retinal precursors, but confirming this hypothesis will require further analysis.

Here, we show that cortical myosin accumulation does not correlate in time with the fast oscillations of the membrane. In spite of this, our data does not allow to rule out a myosin role in the maintenance of the pulsatile state. On the contrary, blebbistatin treatment severely impaired membrane pulses, suggesting that myosin basal activity is necessary to maintain the fast oscillatory behavior. Our data also show that myosin accumulates at the basal cortex in discrete foci, which have an average stability of approximately 4 min and are distributed in scattered cells across the epithelial field. Remarkably, a large proportion of the retinal cells accumulating basal myosin foci are contracting both along the apico-basal and basal plane axes. These episodic contractions at the basal surface can be inhibited either by blocking myosin activity or by interfering with the adhesive properties of the extracellular matrix.

Taken together, our observations suggest a working model for the ratcheted constriction of the epithelium (*Figure 9*). According to this hypothetical model, retinal precursors would experience non-ratcheted fast membrane oscillations. Pulsatile behavior without a net reduction of cell area has also been reported in several epithelial contexts (*He et al., 2010; Solon et al., 2009; Roh-johnson et al., 2012*). Superimposed to these fast oscillations, the episodic accumulation of myosin at the basal surface in scattered cells would mediate their progressive (i.e. ratcheted) constriction. Then, individual contributions would add up over time to cause the constriction of the entire neuroepithelium. At a tissue level, our laser ablation experiments indicate that the global balance between mechanical tensions and tissue resistance becomes transiently unstable within a limited developmental window (19–20 hpf). This critical period, in which local ablations at the basal surface trigger global tissue rearrangement, coincides with the acute bending of the optic cup epithelium and the active constriction of the neuroblasts' feet. Upon *lamc1* knockdown both basal contractility and global tissue response to laser ablation are attenuated. This suggests that the ECM plays a fundamental role in the transmission of mechanical tensions generated by individual cells at the tissue level. In agreement with this concept, previous reports have shown that optic cup morphogenesis largely depends on integrin function (*Martinez-morales et al., 2009; Bogdanovic et al., 2012; Nakano et al., 2012*).

The formation of the eye chamber offers an excellent model to understand basal constriction in epithelia. This study has revealed significant differences in cell and actomyosin dynamics between retinal folding and previously characterized apical constriction processes. To what extent these different features can be attributed to the neuroepithelial character of the retina or are a common theme in epithelial layers undergoing basal constriction remains an open question.

# Materials and methods

## Zebrafish

Adult AB/Tübingen (AB/Tu; RRID:ZIRC_ZL1/RRID:ZIRC_ZL57) wild-type zebrafish strain, transgenic lines *tg(vsx2.2:GFP-caax)* (*Gago-rodrigues et al., 2015*, *tg(actb1:myl12.1-eGFP)* (*Behrndt et al., 2012*), and the mutant strain sleepy (slym86; RRID:ZFIN_ZDB-GENO-090402-2; *Parsons et al., 2002*) were maintained and bred under standard conditions (*Westerfield, 2000*). The line *tg(vsx2.2: lyn-tdTomato)* was generated by recombining the medaka *vsx2.2* promoter (*Martinez-morales et al., 2009*) with the membrane reporter Lyn-tdTomato in the backbone of the destination vector *pDestTol2CG* (*Kwan et al., 2007*). All embryos were staged in hours post-fertilization (hpf) as described (*Kimmel et al., 1995*). All experiments conform national and European Community standards for the use of animals in experimentation.

## Live-imaging

Transgenic embryos were anesthetized using 0.04% MS-222 (Sigma), embedded in 0.8% low-melting agarose in E3 medium, and mounted on 35 mm glass-bottom dishes (WPI-Fluorodish). Time-lapse analyses were performed on a Leica SP5 confocal microscope with a 20x/0.75 IMM multi-immersion objective. Optical sections containing either apical or basal surfaces were identified by z-stacks in resonant mode throughout the entire retinal epithelium (*Figure 1—figure supplement 1*). To determine the orientation of the neuroepithelium along the apico-basal axis and the position of apical and basal surfaces, a z-stack (with 1µm spatial resolution) was taken across the entire retina at the beginning and at the end of each time series. We used this information to establish confocal planes for live imaging 1–3 µm below the surfaces. Then small z-stacks (3 planes over a total of 1 µm) were recorded every 5 or 8 s at the selected planes, 1 µm below the apical or basal surfaces. Long-term recordings along the apico-basal axis were performed using the galvano scanner.

## Image processing and segmentation

Time-lapse images were processed using Fiji (RRID:SCR_002285; *Schindelin et al., 2012*). Different plugins were used for maximum intensity projection of z-stacks, signal intensity quantification in selected regions of interest (ROIs), and measurement of angles and distances. To measure the length of the apical and basal edges of the retina (*Figure 1*), we selected a single stack at the central retina and outlined tissue borders using the Fiji tool freehand. For automatic detection of cell edges and tracking of individual cells through time we used Packing Analyzer v2.0, which is based on a watershed algorithm for cell identification (*Aigouy et al., 2010*). Unique RGB codes were assigned to each cell by Packing Analyzer V2.0 in tracked images. Individual images were examined manually to correct for automatic segmentation mistakes. Only those cells that could be tracked unambiguously through time were considered for quantification (*Figure 1—figure supplement 1*; *Video 13*). Once cell areas were quantified, the constriction rates were calculated as the first derivative of time and represented with Excel (Microsoft) (*Figure 2—figure supplement 1*). For automatic actin intensity measurements (*Figure 4*), individual cell profiles (as revealed by lyn-tdTomato) were segmented and tracked using Packing Analyzer V2.0. This software generates unique RGB codes and masks for every tracked cell. Then, a MATLAB (Mathworks) script was used to overlap cell masks with images showing F-actin (Utrophin-GFP) and to quantify average intensity per cell area.

For cross-correlation analyses of oscillatory signals we use the following equation:

$(f \star g)[n] = F^{-1} \{F \{f^*\} \cdot F \{g\}\}$; where $F^{-1}$ denotes the inverse Fourier transform.

We use the autocorrelation, the cross-correlation of a signal with itself, to normalize the cross-correlation and obtain a cross-correlation coefficient ranging from −1 (maximum inverse correlation) to +1 (maximum correlation).

## Transplantation

Fertilized *Tg(vsx2.2:GFP-caax)* and wild-type eggs were incubated at low density (50 eggs per dish) at 28°C until 4hpf. Then embryos were dechorionated by pronase treatment (375 µg/ml) and gently washed with E3 medium. Cells from the blastula cap of donor embryos were collected with a glass needle (Borosilicate Glass Capillaries GC100-10; 1.0 mm × 58 mm, 6´´. Harvard Apparatus) and implanted into the caps of host embryos. After cell transfers were completed, host and donor

embryos were incubated at 28°C. Once the desired developmental stage is reached (20 hpf), GFP-positive embryos were selected and prepared for in vivo live imaging. Apical and basal oscillations were simultaneously recorded for 10 transplanted neuroblasts from five different retinas.

### RNA injections

To visualize actin dynamic, we used utrophin-GFP as a reporter. The plasmid *pCS2:Utrophin-GFP* (*Burkel et al., 2007*) was used to synthesize the corresponding RNA. The construct was first linearized with NotI (Takara), and RNA was synthesized using the mMESSAGE mMACHINE SP6 kit (Ambion). Capped *utrophin-GFP* RNA was then precipitated with 4M LiCl, quantified, and injected into *Tg(vsx2.2:lyn-tdtomato)* embryos at one-cell stage (200 pg per embryo).

### Lamc1Mo injections

Antisense lamc1morpholino oligonucleotides (MO) were purchased from Gene Tools, LLC. Lamc1Mo 5'-TGTGCCTTTTGCTATTGCGACCTC-3' blocks translation, is complementary to the 5' sequence of *lamc1* and has been shown to phenocopy ocular malformations observed for the *lamc1* mutation *sly* (*Ivanovitch et al., 2013; Parsons et al., 2002*). The lamc1Mo was injected into *tg(vsx2.2:GFP-caax)* and *tg(actb1:myl12.1-eGFP)* embryos at one-cell stage at a concentration of 1 pmol per embryo. To prevent potential apoptotic effects, a p53MO (p53MO: 5'-GCGCCATTGCTTTGCAAGAATTG-3'), was co-injected with Lamc1Mo at a concentration of 0.5 pmoles per embryo. Control embryos were injected in parallel with p53MO alone.

### Laser ablation and spinning disk confocal microscopy

Transgenic embryos were selected at the appropriate developmental stages, dechorionated with forceps, embedded in 0.8% low melting point agarose, and mounted onto 35 mm petri dishes as described above. Embryos were carefully oriented with the dorsal head surface contacting the coverslip and were imaged using a 40x objective. In order to be able to record time-lapse movies with sufficient time resolution (ms) for an optical flow analysis, we used a spinning disk confocal microscope (RoperScientific), achieving a time resolution of 0.5 s for all experiments in this work. Laser ablations were performed by applying a short wavelength laser (405 nm) at single cell membranes for 450 ms, either at the basal or apical surfaces of the neuroretinal tissue. Laser pulses were controlled using iLas software (Roper Scientific). For the statistical analysis of maximal ablation speeds, ablated retinas were sorted in 15° bins.

### Particle flow analysis

In order to assess the retraction speed of the neuroretinal tissue after laser ablation, we measured optical flow between consecutive frames. To compare pixel intensity between frames, we employed the Lucas-Kanade method, which groups neighboring pixels together assuming similar motion for them (*Barron et al., 1994*). The algorithm Good Features to Track was used for the pixel-wise detection of features to track (Shi and Tomasi, 1994). Both methods are available as programming functions at the computer vision open source library, OpenCV (*Bradski and Kaehler, 2008*). Different positions at the central and distal retina and the apical and basal surfaces of the neuro-epithelium were considered for optical flow measurements. For each region, 11 points were tracked and their speed values median-averaged. Retraction speed graphs have been Gaussian smoothed. In order to allow direct comparison between different experiments, speed profiles for each retina analyzed were normalized to their median values.

## Acknowledgements

We thank Elisa Marti, Paola Bovolenta and Caren Norden for their critical input and to Javier Montaño for his advice on Matlab. We are in debt to Ana Fernández-Miñán (Aquatic Vertebrates Platform) for all the help with transgenesis and transplantation experiments, and to Katherina García for her excellent technical assistance in the imaging facility. The authors wish to thank the financial support given to M N-P by the FPI-MICINN program. This work was supported by grants *BFU2011-22916, P11-CVI-7256, BFU2014-53765 and BFU2014-55738-REDT* to JRMM.

## Additional information

### Funding

| Funder | Grant reference number | Author |
|---|---|---|
| Ministerio de Economía y Competitividad | BFU2011-22916 | Juan R Martínez-Morales |
| Ministerio de Economía y Competitividad | P11-CVI-7256 | Juan R Martínez-Morales |
| Ministerio de Economía y Competitividad | BFU2014-53765 | Juan R Martínez-Morales |
| Ministerio de Economía y Competitividad | BFU2014-55738-REDT | Juan R Martínez-Morales |

The funders had no role in study design, data collection and interpretation, or the decision to submit the work for publication.

### Author contributions

MN-P, JRM-M, Conception and design, Acquisition of data, Analysis and interpretation of data, Drafting or revising the article; FK, JL, RP, Conception and design, Acquisition of data, Analysis and interpretation of data; JW, Conception and design, Analysis and interpretation of data, Drafting or revising the article

### Author ORCIDs

Jochen Wittbrodt, http://orcid.org/0000-0001-8550-7377
Juan R Martínez-Morales, http://orcid.org/0000-0002-4650-4293

### Ethics

Animal experimentation: All experiments conform national (RD53/2013) and European Community standards for the use of zebrafish in experimentation. This work has been approved by three independent comittees on the Ethics of Animal Experiments at the Pablo de Olavide University, the National Reseach Council (CSIC) and the local Goverment of Andalucia (permit number 26-11-14-164).

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
