## [Decision Letter]

Thank you for submitting your article "Analysis of cellular behavior and cytoskeletal dynamics reveal a ratchet-like mechanism driving optic cup morphogenesis" for consideration by *eLife*. Your article has been reviewed by three peer reviewers, one of whom Suzanne Eaton (Reviewer #1), is a member of our Board of Reviewing Editors, and the evaluation has been overseen by Janet Rossant as the Senior Editor.

The reviewers have discussed the reviews with one another and the Reviewing Editor has drafted this decision to help you prepare a revised submission.

The authors have made many interesting and novel observations and the data generally support a model in which laminin- and myosin-dependent basal constriction contributes to morphogenesis of the optic cup. However the reviewers agreed that many aspects of the quantitative analysis need to be explained in more detail before the paper would be suitable for publication. These are outlined below. One concern was that it was not clear from how many different embryos the data was derived. Of course it is important that these results are consistent over multiple embryos, and if only one has been examined for each type of experiment, then additional data needs to be provided from several others. Otherwise, no additional experiments are needed. All reviewers were unsatisfied with the use of the term "rachet-like" mechanism and thought the data didn't necessarily support such a specific description – this term should be removed from the title, although it would be reasonable to raise the possibility in the discussion.

Improvement to explanation and presentation of data:

1) The authors perform laser ablation experiments to look at tension at the basal side over time, both in wild type and in lamc morphants. They suggest that basal tension is dependent on lamc and is highest at the time that cells are contracting the most – at this time, ablation can affect the global shape of the cup. But Figure 8, which presents this data, is very hard to understand. In particular, the direction that the cells move after laser ablation is impossible to see. The image has been color coded in a way that is said to represent "motion vectors", but it isn't clear whether the color represents the magnitude or the direction of the vectors. Showing a velocity flow field for the tissue would be a better way of presenting the global effects of laser ablation, and it is essential to show a higher magnification view of the cut site itself so that it is clear that tissue retracts after cutting.

2) Most of the authors' experiments rely on rapid imaging of small sets of z-slices near the apical and basal surfaces of the neuroepithelium. It is difficult to tell from Figure 1—figure supplement 1 how the authors can tell if they are imaging exactly at the basal surface. A slight tilt in the axis of the eye or embryo would skew the precise measurements being taken: the area would not represent the basal surface, but instead, an oblique section nearby. Can the authors clarify their methods to state how z-slices were selected and validated?

3) Figure 1, apical and basal length measurements: Can the authors add a short description to the methods section as to how this was carried out? How was the single optical section within the z-stack selected for the measurement?

4) Results section and Figure 1: "Cell elongation[…]does not occur during retinal folding". I find this comment a bit confusing. These measurements were performed between 19 hpf and 22 hpf. But as shown in Figure 1, some amount of retinal folding seems to occur between 17 and 19 hpf. Can the authors clarify how they defined the period of retinal folding?

5) In the same section: "basal areas shrank significantly (40%) and irreversibly[…]" Can the authors clarify the evidence (particularly at this point in the manuscript) for irreversibility of the phenomenon?

6) Figure 1—figure supplement 2: The authors note that for each timepoint, 24 cells were monitored. Were these the same 24 cells at each timepoint? How many embryos were used for these measurements?

7) Figure 3 understand why the authors might have chosen to move to length measurements here, as opposed to the area measurements in Figure 2. However, I have concerns about how this was done: are the images in Figure 3 (and Video 3) z-projections to ensure that the entire depth of the cells (and therefore width) can be accounted for? In addition, how does the variance in length compare to the variance seen in the area measurements in Figure 2? Was it not possible to acquire z-stacks to measure the 2-dimensional area of apical and basal surfaces, perhaps after 3-dimensional rendering? Finally, in Video 3, there appears to be a protrusion at the basal end of the cell – was this taken into account for the analysis?

8) Figure 3—figure supplement 1: Were these mitotic cell measurements performed within a single optical section? Is it possible that there was displacement of neighboring cells in the z-axis?

9) Figure 8: Can the authors provide supplemental images to demonstrate how angles of invagination were measured? I am having a hard time seeing 120 degrees in B and 45 degrees in C.

10) Figure 8: How were the regions selected for optical flow quantification? A representative image at each age with central and distal points marked would be helpful in understanding the quantifications here.

11) The data presented show very little statistics. In the Figure 2, only 3 cells are represented and a couple of cells in supplemental figure. Similar observation can be made on the Figure 3 were only quantifications for 1 clone are shown or for the Figure 6 on the correlation between axis shortening and myosin accumulation. The authors should increase the number of cells analyzed in these figures, (for example in supplemental figures).

12) The oscillations observed and quantified by the authors have rather small amplitude. It is difficult to identify the osciilations on the videos and to evaluate the precision of the segmentation. The authors should modify the suppl. Video showing apical and basal oscillations to include also the segmented cell. In that way we could evaluate the accuracy of the segmentation and measurements.

13) About the Figure 4 and the correlations between actin levels and cell area. The details of the analysis of the actin levels are missing. It is difficult to assess exactly what the authors are measuring. For instance, I wonder whether the fluctuations in actin levels are not coming to changes in cell perimeter or perimeter to surface ratio. The authors should clearly explain the methodology used to extract fluorescence levels.

14) In the cases of Blebbistatin and laminin morpholino experiments, quantification of basal or apical cell surface areas are missing. The authors should show the effect of these treatments on the oscillations.

Changes to the text/discussion:

Discussion section: "We then examined[…] where the irreversible constriction takes place." As noted above, can the authors clarify the evidence for irreversibility? Is the constriction irreversible if embryos are manipulated in a different way (e.g. with latrunculin to depolymerize actin filaments)?

Subsection “Analysis of tension distribution during optic cup morphogenesis by laser ablation.2 and Video 11: "Tissue ablations at the apical surface did not seem to affect the global geometry of the retina." It seems to me that Video 11 shows that upon apical ablation, the retina appears to recoil toward the basal side, causing some amount of retinal flattening; this seems to be similar to the way that the basal retina recoils toward the apical side upon basal ablation, though basal ablation leads to the opposite effect (increased folding). This suggests that the apical surface is under tension as well, which is interesting. Additionally, the prospective retinal pigmented epithelium is in close proximity to the apical surface of the neural retina: are those cells being ablated during the apical ablation?

In the first sentence of the Abstract the authors mention: "Tissue morphogenesis depends on the dynamic flow of contractile actomyosin networks". This is too strong. Some morphogenetic rearrangements associate with actomyosin flow but this is far to be clear that epithelial morphogenesis depends on actomyosin flow.

[Editors' note: further revisions were requested prior to acceptance, as described below.]

Thank you for resubmitting your work entitled "Analysis of cellular behavior and cytoskeletal dynamics reveal a constriction mechanism driving optic cup morphogenesis" for further consideration at *eLife*. Your revised article has been favorably evaluated by Janet Rossant (Senior editor) and two reviewers.

The manuscript has been improved but there are some remaining issues that need to be addressed before acceptance, as outlined below by the reviewers.

Reviewer #2:

et al.The authors' revised manuscript addresses most of my concerns raised during initial review, and the revisions and clarifications strengthen their conclusions. I still have a few minor comments to be addressed, simply related to further clarifications required and concerns about statements made in the text.

1) In the rebuttal #13 (Figure 4), the authors have clarified for me how the actin quantification was done. However, I feel that the wording in the methods section is still very general, and when reading through the methods, it was still not clear to me that this was actually the procedure used to quantify actin signal, and that it was quantified averaged per unit cell area (if I am interpreting the authors correctly). I would request that the authors please add a note of clarification to the methods that this was used for Figure 4 and how quantifications specific to Figure 4 were performed.

2) Results section: "[…]whereas the active oscillatory behavior at the basal side resulted in a progressive reduction of cell area[…]" At this point in the manuscript, the authors have not shown that oscillatory behavior is causative, merely that it correlates with reduction in cell area.

3) Results section: "[…] suggesting than [sic] actin needs to accumulate over a threshold to have an effect on cell size." This statement suggests a causal relationship between actin accumulation and cell size, and such a relationship has not been shown; this is correlative. Experiments in which actin depolymerizing agents are used would help to establish that relationship.

4) Figure 7—figure supplement 1, panel D: the statistical comparisons being made in the graph are confusing to me; it is not clear from the positioning of the asterisks which comparisons are being considered statistically significant. A simple repositioning of the asterisks would help here.

5) Subsection “*Lamc1 function is required for efficient cell contractility, basal constriction and optic cup folding”*: the authors refer to embryos injected with lamc1 morpholino as "adhesion-deficient embryos". The authors have not shown that adhesion itself is deficient, only that lamc1 is likely knocked down.

6) Discussion, paragraph four: I feel that the proposal of this working model would benefit from a model diagram to accompany it, possibly comparing wild type and lamc1 morphant oscillations and constriction.

Reviewer #3:

The authors have significantly improved the quality of the manuscript. Particularly on the clarity of the statistics and methodology. I think the manuscript is now close to being acceptable for publication in *eLife*. I have however few comments that would be necessary to be implemented before acceptation.

Specific comments:

1) The authors have now given epithelial width at earlier times (17hpf) in the answers to reviewers but not included it in the manuscript. I think these data should be included in the final version.

2) About actin intensity measurements on the Figure 4, The methodology is still insufficiently explained. We observe normalized Utrophin-GFP fluctuating between 5 and -5. I understand from the explanation that these are proportional to mean pixel intensity over the entire surface of each segmented cell. Is it correct? How is it normalized to get positive and negative values? It should be more explicitely explained in the methods.

3) On the same Figure 4, it is labelled (u.a) where it should be (a.u.) for arbitrary units. Also How the cell area rate is normalized? It is not explicitly mentioned and depending on the normalization the units may not be arbitrary.

4) On the Figure 7—figure supplement 1 C the Avg peak amplitude units is microns2/min. I guess it is microns2. Also the peak amplitude seems to scale with the cell size of LamC morphants that is larger. This could be included in the text.

---

## [Author Response]

[…]

*Improvement to explanation and presentation of data:*

*1) The authors perform laser ablation experiments to look at tension at the basal side over time, both in wild type and in lamc morphants. They suggest that basal tension is dependent on lamc and is highest at the time that cells are contracting the most – at this time, ablation can affect the global shape of the cup. But Figure 8, which presents this data, is very hard to understand. In particular, the direction that the cells move after laser ablation is impossible to see. The image has been color coded in a way that is said to represent "motion vectors", but it isn't clear whether the color represents the magnitude or the direction of the vectors. Showing a velocity flow field for the tissue would be a better way of presenting the global effects of laser ablation, and it is essential to show a higher magnification view of the cut site itself so that it is clear that tissue retracts after cutting.*

We have made an effort to improve the quality of the overlays between the membrane signal and the particle tracking analysis in the ablation experiments. This has improved the signal to noise ratio in panels A-C in Figure 8 as well as in Video 12. In addition a new supplementary figure (Figure 8—figure supplement 1) has been generated to make clear the color code used for particle tracking: i.e. different colors corresponding to the direction of the displacement and color intensity to the magnitude of the displacement (this explanatory sentence is now also included in the figure legends). Higher magnification views of the cut sites are also provided in this new supplementary figure.

We are confident that the updated version of the Figure 8 and the Video 12, as well as the new supplementary figure will be sufficient to answer all the concerns on this issue.

*2) Most of the authors' experiments rely on rapid imaging of small sets of z-slices near the apical and basal surfaces of the neuroepithelium. It is difficult to tell from Figure 1—figure supplement 1 how the authors can tell if they are imaging exactly at the basal surface. A slight tilt in the axis of the eye or embryo would skew the precise measurements being taken: the area would not represent the basal surface, but instead, an oblique section nearby. Can the authors clarify their methods to state how z-slices were selected and validated?*

To determine the orientation of the retinal neuroepithelium along the apico-basal axis and the position of apical and basal surfaces, a z-stack (with 1µm spatial resolution) was taken across the entire retina at the beginning and at the end of each time series (similarly to what is shown in Figure 1—figure supplement 1). We used this information to establish confocal planes for imaging between 1-3 µm below the apical or basal surfaces. (Note: the size and regular geometry of the retinal primordium, together with its relatively smooth surfaces (Figure 1), facilitate the identification of the terminal planes).

To clarify this point, we have now included the following information in the paragraph of the methods section live-imaging: “Optical sections containing either apical or basal surfaces were identified by z-stacks in resonant mode throughout the entire retinal epithelium (Figure 1—figure supplement 1). […]Then small z-stacks (3 planes over a total of 1 µm) were recorded every 5 or 8 seconds at the selected planes”

*3) Figure 1, apical and basal length measurements: Can the authors add a short description to the methods section as to how this was carried out? How was the single optical section within the z-stack selected for the measurement?*

Apical and basal edges indicated in Figure 1 with dotted lines were measured using the tool freehand line (Fiji) to outline the retinal borders. For these measurements an optical plane was selected at the central retina using as a reference the maximum lens vesicle diameter. To clarify this point, we have now included the information in the methods section Imaging processing and segmentation: “To measure the length of the apical and basal edges of the retina (Figure 1) we selected a single stack at the central retina and outlined tissue borders using the Fiji tool freehand”

*4) Results section and Figure 1: "Cell elongation[…]does not occur during retinal folding". I find this comment a bit confusing. These measurements were performed between 19 hpf and 22 hpf. But as shown in Figure 1, some amount of retinal folding seems to occur between 17 and 19 hpf. Can the authors clarify how they defined the period of retinal folding?*

The complete epithelialization of the retinal precursors is achieved around 16-17 hpf, when all the neuroblasts orient their a-b axis towards the lens primordium (Kwan et al. 2012, Ivanovitch et al. 2013, and our own observations). We define the retinal folding period from this moment until 22-23 hpf when optic cup morphogenesis is completed; as included in Figure 1. Our observations indicate that the width of the retinal epithelium remains invariant throughout all this process, spanning approximately 50 µm. It is true that the measurements provided in Figure 1 correspond only to time-lapse series between 19 hpf and 22 hpf. These measurements derive from three long recordings taken exclusively for that purpose with identical confocal settings.

To rule out changes in epithelial width at earlier stages we have performed new measurements at 17 hpf from 6 videos. The obtained values for the anterior (49.6 ± 3.5 µm), central (49.9 ± 5.3 µm), and posterior (47.6 ± 5.5 µm) retinas are not significantly different from that previously obtained for older stages (n=6; T-test) (See Figure 1). These results confirm that the width of the retinal epithelium remains invariant throughout optic cup folding. Because the new 17 hpf measurements do not belong to the same recording series included in Figure 1, and in any case they do not modify our previous conclusions, we have decided to maintain the figure in its current format.

*5) In the same section: "basal areas shrank significantly (40%) and irreversibly…" Can the authors clarify the evidence (particularly at this point in the manuscript) for irreversibility of the phenomenon?*

We do agree with the reviewers. Claiming that the constriction phenomenon is irreversible may be an unnecessary overstatement not sufficiently supported by the data. Thus, we have substituted in the manuscript the terms “irreversibility” and “irreversible” by the more descriptive terms “progressively” and “progressive”.

*6) Figure 1—figure supplement 2: The authors note that for each timepoint, 24 cells were monitored. Were these the same 24 cells at each timepoint? How many embryos were used for these measurements?*

For the experiment in Figure 1—figure supplement 2 we recorded a total of 24 cells from three different embryos, either at the apical or at the basal side (i.e. a total of six independent embryos were recorded). The same eyes were monitored at the central retina through time (in fact, for each time point we recorded 30 min videos). Focal planes needed to be adjusted every 30 min to guarantee that we were still recording 1-3 µm below the apical or basal surfaces. To make this point clear in the text, we have included the following information in the figure legend: “A total of 24 cells from three different embryos were recorded either at the apical or at the basal side.”

*7) Figure 3 understand why the authors might have chosen to move to length measurements here, as opposed to the area measurements in Figure 2. However, I have concerns about how this was done: are the images in Figure 3 (and Video 3) z-projections to ensure that the entire depth of the cells (and therefore width) can be accounted for? In addition, how does the variance in length compare to the variance seen in the area measurements in Figure 2? Was it not possible to acquire z-stacks to measure the 2-dimensional area of apical and basal surfaces, perhaps after 3-dimensional rendering? Finally, in Video 3, there appears to be a protrusion at the basal end of the cell – was this taken into account for the analysis?*

The objective of this experiment was to investigate the existence of two possible mechanisms:

a) The occurrence of coordinated membrane pulses transmitted as waves across the entire apico-basal axis of the cell.

b) The coordinated contraction of apical and basal surfaces either simultaneously or asynchronously.

Our results, derived from the observation of 10 transplanted neuroblasts (from 5 different retinas), do not support any of these possibilities, but rather indicate that apical and basal surfaces behave as independent oscillators.

We do agree with the reviewer in that 3D renderings of the cells may have allowed reconstructing the apical and basal surfaces. In fact, we tried this approach on the transplanted neuroblasts. Unfortunately, this turned to be problematic, as it was technically difficult to keep the high-temporal and spatial resolution necessary while recording enough z-planes for a high quality 3D rendering. As an alternative approach we decided to measure the cell diameter in a maximum projection of 3 z-stacks (over a total of 1µm) selected at the maximum width of the cell (as determine after a preliminary z-stack reconstruction). This information is already included in the legend of Video 3. Since the measured parameter was the basal end diameter, cell protrusions were not taken into account in our analysis.

Regarding the question of as to how does the variance in length compare to the variance in area: a geometric calculation shows that a typical variation of 10% in cell area implies a diameter variation of ≈ 5%, which still is within the resolution range of our videos. Therefore, we believe that the measurement of diameter variations is a valid approach to answer the intended question.

*8) Figure 3—figure supplement 1: Were these mitotic cell measurements performed within a single optical section? Is it possible that there was displacement of neighboring cells in the z-axis?*

Mitotic cells measurements in Figure 3—figure supplement 1 were obtained for a total of 10 cells from 3 different retinae. Measurements were taken from maximum projections of 3 z-stacks (over a total of 1µm). This does not allow ruling out whether there is neighboring cells displacement in the z-axis. To address this question we examined 5 different cell divisions (from different retinas) at an apical plane. The results confirm our previous observations indicating that apical expansion along the mitotic axis is transient. In addition neighbor relationships remain approximately constant during cell divisions, thus indicating that apical expansion does not occur either perpendicularly to the mitotic axis. We have included these results in Figure 3—figure supplement 1.

*9) Figure 8: Can the authors provide supplemental images to demonstrate how angles of invagination were measured? I am having a hard time seeing 120 degrees in B and 45 degrees in C.*

We thank the reviewers for bringing our attention to this point. We have realized that the criteria used for angles measurement in Figure 8 (i.e. setting the angle vertex at the center of the retina epithelium) was different from that used in the rest of the figures (i.e. in Figure 6 and Figure 7 the angle vertex was anchored at the basal surface). We apologize for this inconsistent protocol, which in any case do not change our conclusions (i.e. on the existence of a critical period, in which local ablations at the basal surface trigger global tissue rearrangement).

To resolve this issue we have measured again all the retinal angles in Figure 8, associated supplementary figures, and Video 10, Video 11 and Video 12, anchoring now the vertex to the basal surface. Angle measurements have been corrected accordingly in the figures (120º and 45º are now 130º and 80º respectively), and bending angles are now indicated with dashed lines. In addition we have also realized that angles nomenclature in Figure 8 was not sufficiently explained. For the statistical analysis of maximal ablation speeds, ablated retinas were sorted in 15º bins. This is now described in the laser ablation section in methods and the exact bins indicated in the corresponding box plots.

*10) Figure 8: How were the regions selected for optical flow quantification? A representative image at each age with central and distal points marked would be helpful in understanding the quantifications here.*

Boxes showing the regions selected for optical flow quantification are now indicated in Figure 8—figure supplement 1.

*11) The data presented show very little statistics. In the Figure 2, only 3 cells are represented and a couple of cells in supplemental figure. Similar observation can be made on the Figure 3 were only quantifications for 1 clone are shown or for the Figure 6 on the correlation between axis shortening and myosin accumulation. The authors should increase the number of cells analyzed in these figures, (for example in supplemental figures).*

We believe that in general our observations have enough statistical support. In the revised version we have scanned the manuscript to make sure this is properly stated in the text when necessary. Nevertheless, we have revisited Figure 2, Figure 3 and Figure 6 as follow:

Figure 2: We are showing apical and basal oscillations for 6 representative cells in Figure 2 and Figure 2—figure supplement 1. However our conclusions are based on the recording of 43 individual cells at the apical side and 46 cells at the basal side. In both cases, cell oscillations were examined in three independent retinas. In addition, we have provided the average frequency and peak-to-peak amplitude in the text accompanying Figure 2, and the average area through time in Figure 1—figure supplement 2. Most of this information was partially indicated in the text:

“more than 76% of the apical and 90% of the basal oscillations analyzed (n=43) presented no major correlation with those of their neighbors (Pearson correlation coefficient R < |0.5|). Comparison of the pulsatile behavior at both epithelial planes revealed significant differences. Although both surfaces oscillate with a similar frequency of 50 ± 12.5 mHz (≈ 20 ± 5 sec.; n=26 cells), the pulsing amplitude * is considerably larger at the basal 11.1 ± 1,3 µm2/min than at the apical surface 4.1 ± 0.57 µm2/min (Figure 2—figure supplement 1)”.

*Note: we realized that the term “pulsing amplitude” as it was included in page 7 was misleading as it was actually referring to “peak-to-peak amplitude”. We have reserved the term amplitude for semi-amplitudes in other sections of the manuscript.

We have now expanded this description in the first sentence to make the point clear:

“The analysis of individual cells from three independent retinas revealed that 76% of the apical (n=43) and 90% of the basal (n=46) oscillations presented no major correlation with those of their neighbors (Pearson correlation coefficient R < |0.5|).”

In addition we have added new panels to Figure 2—figure supplement 1 (panels E, F, and G) showing the distribution of the correlation coefficients between the oscillations of cell pairs at the apical and basal surface. We strongly believe that these analyses should be sufficient to evaluate the accuracy of the observations included in Figure 2. We think that all this information will be more informative than showing additional examples of the apical and basal oscillations, such as those included in Figure 2; the new Figure 7—figure supplement 1, and the panels included in Figure 10 and Figure 11.

Author response image 1.**DOI:**
http://dx.doi.org/10.7554/eLife.15797.036

Author response image 2.**DOI:**
http://dx.doi.org/10.7554/eLife.15797.037

In Figure 3 we included a representative example of the simultaneous recording of apical and basal oscillations in transplanted cells. As mentioned in the text and in the point 7 of this reply letter, our conclusions are derived from the observation of 10 transplanted neuroblasts from 5 different retinas (the last is now indicated in the figure legend and the methods section). To make this more evident in Figure 3, we have now included a box plot showing the distribution of the correlation coefficients for the apical vs. basal surface in the 10 clones analyzed (Figure 3). We believe that this graphic will be more informative than showing another example of the correlative oscillations, such as the one we have included in Figure 11.

Figure 6: The data presented in this figure are important for the general conclusions of the work. Therefore, in line with the reviewers’ comments, we have carried out new imaging experiment to increase the number of cells analyzed; in Figure 11 20 cells from at least 10 different retinas in each type of experiment. In addition, we are showing additional examples of cellular behavior in the modified Figure 6 and the new supplementary figure (Figure 6—figure supplement 1). Furthermore, we are now including quantitative data on of axial cell shortening (µm) and basal area contraction (µm^2^) upon accumulation of myosin foci: Now in Figure 6 and Figure 6—figure supplement 1, respectively.

We believe that these new analyses strongly support our conclusions regarding the correlation between myosin foci accumulation and basal feet constriction.

*12) The oscillations observed and quantified by the authors have rather small amplitude. It is difficult to identify the osciilations on the videos and to evaluate the precision of the segmentation. The authors should modify the suppl. Video showing apical and basal oscillations to include also the segmented cell. In that way we could evaluate the accuracy of the segmentation and measurements.*

As we mentioned in the text the peak-to-peak amplitude at the basal and apical surfaces are 11.1 ± 1,3 μm^2^/min and 4.1 ± 0.57 μm^2^/min, respectively. This, for basal oscillations, corresponds to surface area changes varying between 10 to 15% of the total area, which is within the range of what has been observed for basal oscillations in *Drosophila* follicle cells (He et al. 2010).

Regarding the methodological issue on the evaluation of the accuracy of the segmentation and measurements, we have now included a new video (Video 13) in which membrane oscillations and automatic cell edge detection are shown in parallel. This video shows the RGB codes assigned by the segmentation program to each tracked cell.

In addition, although we have stated in the methods (imaging processing and segmentation) that: “Automatic detection of cell edges and tracking of individual cells through time was performed with Packing Analyzer v2.0, which uses a watershed algorithm for cell identification (Aigouy et al., 2010). Individual images were examined manually to correct for automatic segmentation mistakes”. Now we have added the following explanatory sentence to the methods section: “Unique RGB codes were assigned to each cell by Packing Analyzer V2.0 in tracked images. Only those cells that could be tracked unambiguously through time were considered for quantification.”

*13) About the Figure 4 and the correlations between actin levels and cell area. The details of the analysis of the actin levels are missing. It is difficult to assess exactly what the authors are measuring. For instance, I wonder whether the fluctuations in actin levels are not coming to changes in cell perimeter or perimeter to surface ratio. The authors should clearly explain the methodology used to extract fluorescence levels.*

Average rather than total pixel intensity was considered for the quantification of actin levels. Thus, fluctuations in actin levels do not come from changes in cell area, but from actin accumulation. We stated this already in the methods section -“For automatic intensity measurements, segmented and tracked images were processed using a MATLAB (Mathworks) script for mean pixel intensity quantification". However, we have decided to explain this point better in the revised version and thus the sentence has been modified as follow: "For automatic intensity measurements, segmented images and RGB codes assigned by Packing Analyzer V2.0 were used to quantify mean pixel intensity for each tracked cell using a script in MATLAB (Mathworks)."

*14) In the cases of Blebbistatin and laminin morpholino experiments, quantification of basal or apical cell surface areas are missing. The authors should show the effect of these treatments on the oscillations.*

We have partially addressed this issue in Figure 6—figure supplement 1 (and Video 8) in which we compare basal oscillations, average peak amplitude and basal constriction in control and blebbistatin treated embryos. To show the effect of *lamc1* loss of function on the cell oscillations we have carried out new imaging experiments. The results obtained are summarized in a new supplementary figure (Figure 7—figure supplement 1). This analysis shows that basal oscillations are not impaired upon *lamc1* knockdown; on the contrary, the average peak amplitude is significantly increased in the morphant cells by 45%. Interestingly, constriction of the cell feet is severally impaired in *lamc1*Mo embryos and the basal cell areas are significantly larger when compared to the control situation (Figure 7—figure supplement 1). This is an interesting observation that is in line with our previous work (Martinez-Morales et al. 2009; Bogdanovic et al. 2012) and that confirms (through a direct observation) that the constriction process requires laminin-dependent adhesion to the ECM. We thank the referee for suggesting this informative experiment.

*Changes to the text/discussion:*

*Discussion section: "We then examined… where the irreversible constriction takes place." As noted above, can the authors clarify the evidence for irreversibility? Is the constriction irreversible if embryos are manipulated in a different way (e.g. with latrunculin to depolymerize actin filaments)?*

This comment is in line with point 5 (see above). What we observe through development is a progressive reduction of the cell areas at the basal surface. We agree with the reviewers in that the term “irreversible” is loaded with mechanistic implications not sufficiently supported by the data. We think that an experimental demonstration for irreversibility may be extremely difficult and goes beyond the scope of this work. Thus, as stated in point 5, we have substituted in the manuscript the terms “irreversibility” and “irreversible” by the more descriptive terms “progressively” and “progressive”.

*Subsection “Analysis of tension distribution during optic cup morphogenesis by laser ablation.2 and Video 11: "Tissue ablations at the apical surface did not seem to affect the global geometry of the retina." It seems to me that Video 11 shows that upon apical ablation, the retina appears to recoil toward the basal side, causing some amount of retinal flattening; this seems to be similar to the way that the basal retina recoils toward the apical side upon basal ablation, though basal ablation leads to the opposite effect (increased folding). This suggests that the apical surface is under tension as well, which is interesting. Additionally, the prospective retinal pigmented epithelium is in close proximity to the apical surface of the neural retina: are those cells being ablated during the apical ablation?*

Tissue reaction to basal vs. apical laser ablation is shown in Video 11 and both central and peripheral tissue displacements are indicated with green and white arrowheads respectively. We do agree in that a response is also observed upon apical ablation, which indicates that this surface is also under tension. It is likely that the apical actomyosin belt observed in the neural retina (Video 4) plays a role in maintaining this tension. In any case, we clearly observed a differential behaviour between apical and basal ablations. Whereas basal ablations resulted in a global tissue displacement affecting not only the central but also the peripheral retina, apical ablations only affected the morphology of the central retina and no peripheral displacement was observed (white arrowhead in Video 11). To make this point clear we have substituted the sentence "Tissue ablations at the apical surface did not seem to affect the global geometry of the retina" by the more explanatory sentence “In contrast to the global reaction observed upon basal ablation, which triggers the displacement of the peripheral retina, apical ablations only affected the morphology of the central retina but no peripheral retraction was observed (Video 11). Furthermore, we have generated a new supplementary figure showing measurements for central and peripheral displacements after ablation (Figure 8—figure supplement 2).

Regarding the RPE issue, we cannot rule out a possible contribution from this tissue to apical tension. Although laser ablations were always aimed at the neural retina membranes, we may have unintentionally damaged the pigmented tissue in some experiments. However, since the main conclusions of the article are related to retinal behaviour upon basal ablation, we have not explored this interesting possibility further.

*In the first sentence of the Abstract the authors mention: "Tissue morphogenesis depends on the dynamic flow of contractile actomyosin networks". This is too strong. Some morphogenetic rearrangements associate with actomyosin flow but this is far to be clear that epithelial morphogenesis depends on actomyosin flow.*

We agree with the point. The sentence has been modified as follow: “Contractile actomyosin networks have been shown to power tissue morphogenesis.”

[Editors' note: further revisions were requested prior to acceptance, as described below.]

[…]

*Reviewer #2:*

*The authors' revised manuscript addresses most of my concerns raised during initial review, and the revisions and clarifications strengthen their conclusions. I still have a few minor comments to be addressed, simply related to further clarifications required and concerns about statements made in the text.*

*1) In the rebuttal #13 (Figure 4), the authors have clarified for me how the actin quantification was done. However, I feel that the wording in the methods section is still very general, and when reading through the methods, it was still not clear to me that this was actually the procedure used to quantify actin signal, and that it was quantified averaged per unit cell area (if I am interpreting the authors correctly). I would request that the authors please add a note of clarification to the methods that this was used for Figure 4 and how quantifications specific to Figure 4 were performed.*

Following the reviewer’s suggestion we have include additional information on the quantification of actin intensity in the methods section. The following sentence “For automatic intensity measurements, segmented images and RGB codes assigned by Packing Analyzer V2.0 were used to quantify mean pixel intensity for each tracked cell using a script in MATLAB (Mathworks).” has now being replaced by the more explanatory paragraph:

“For automatic actin intensity measurements (Figure 4), individual cell profiles (as revealed by lyn-tdTomato) were segmented and tracked using Packing Analyzer V2.0. This software generates unique RGB codes and masks for every tracked cell. Then, a MATLAB (Mathworks) script was used to overlap cell masks with images showing F-actin (Utrophin-GFP) and to quantify average intensity per cell area.”

*2) Results section: "[…]whereas the active oscillatory behavior at the basal side resulted in a progressive reduction of cell area[…]" At this point in the manuscript, the authors have not shown that oscillatory behavior is causative, merely that it correlates with reduction in cell area.*

The reviewer is right. We have changed the sentence as follows: "[…]whereas a progressive reduction of cell area was apparent at the basal side, cells did not display a net constriction at the apical side over a 25 min period (Figure 2).”

*3) Results section: "[…]suggesting than [sic] actin needs to accumulate over a threshold to have an effect on cell size." This statement suggests a causal relationship between actin accumulation and cell size, and such a relationship has not been shown; this is correlative. Experiments in which actin depolymerizing agents are used would help to establish that relationship.*

We agree with the referee in that we have not formally demonstrated a causal relationship. Regarding the use of actin-depolymerizing drugs, such as cytochalasin B, we were concerned about that their general cytotoxic effects may obscure the interpretation of the experiments. We believe that addressing causality between actin accumulation and cell feet size would require interfering locally with actin polymerization. Since we think these are complex experiments beyond the scope of this work, we have decided to simply remove this sentence from the final version.

*4) Figure 7—figure supplement 1, panel D: the statistical comparisons being made in the graph are confusing to me; it is not clear from the positioning of the asterisks which comparisons are being considered statistically significant. A simple repositioning of the asterisks would help here.*

The asterisks have been repositioned to clarify this point.

*5) Subsection “Lamc1 function is required for efficient cell contractility, basal constriction and optic cup folding”: the authors refer to embryos injected with lamc1 morpholino as "adhesion-deficient embryos". The authors have not shown that adhesion itself is deficient, only that lamc1 is likely knocked down.*

We have modified this sentence as follow: “To investigate myosin dynamics in the folding retina of lamc1-deficient embryos, morpholinos were injected[…]”

*6) Discussion, paragraph four: I feel that the proposal of this working model would benefit from a model diagram to accompany it, possibly comparing wild type and lamc1 morphant oscillations and constriction.*

A new figure (Figure 9) has been generated to illustrate the working model suggested in the Discussion.

*Reviewer #3:*

*The authors have significantly improved the quality of the manuscript. Particularly on the clarity of the statistics and methodology. I think the manuscript is now close to being acceptable for publication in eLife. I have however few comments that would be necessary to be implemented before acceptation.*

*Specific comments:*

*1) The authors have now given epithelial width at earlier times (17hpf) in the answers to reviewers but not included it in the manuscript. I think these data should be included in the final version.*

These data have now been included in Figure 1.

*2) About actin intensity measurements on the Figure 4, The methodology is still insufficiently explained. We observe normalized Utrophin-GFP fluctuating between 5 and -5. I understand from the explanation that these are proportional to mean pixel intensity over the entire surface of each segmented cell. Is it correct?*

Yes, that is correct. Please see response to reviewer #2. We have already included more methodological information in the revised version

*How is it normalized to get positive and negative values? It should be more explicitely explained in the methods.*

Please see below.

*3) On the same Figure 4, it is labelled (u.a) where it should be (a.u.) for arbitrary units. Also How the cell area rate is normalized? It is not explicitly mentioned and depending on the normalization the units may not be arbitrary.*

In Figure 4 we normalized the area rate and the Utrophin-gfp rates dividing by the mean of their absolute values. Thus, a.u. (the labels have been now corrected in Figure 4) is justified in the legends. The fact that both rates fluctuate between positive and negative values simply reflects the relaxation/contraction of cell area and the concentration/dilution of actin levels. We have now included this information in the corresponding figure legend.

*4) On the Figure 7—figure supplement 1 C the Avg peak amplitude units is microns2/min. I guess it is microns2. Also the peak amplitude seems to scale with the cell size of LamC morphants that is larger. This could be included in the text.*

In the panel C we are representing the average peak amplitude of the cell area rate (in µm2/min). This however was not properly mentioned. We have now included this information in the figure legend.